# Associations Between Follicular Fluid Biomarkers and IVF/ICSI Outcomes in Normo-Ovulatory Women—A Systematic Review

**DOI:** 10.3390/biom15030443

**Published:** 2025-03-20

**Authors:** Soha Albeitawi, Saif-Ulislam Bani-Mousa, Baraa Jarrar, Ibrahim Aloqaily, Nour Al-Shlool, Ghaida Alsheyab, Ahmad Kassab, Baha’a Qawasmi, Abdalrahman Awaisheh

**Affiliations:** 1Department of Pediatrics, Family Medicine and Obstetrics & Gynecology, Faculty of Medicine, Yarmouk University, Irbid 21163, Jordan; bara.khalil33@hotmail.com (B.J.); ibrahim.aloqaily76@gmail.com (I.A.); noorshlool72@gmail.com (N.A.-S.); ghaidaalsheyab@gmail.com (G.A.); kassabahmad628@gmail.com (A.K.); bahaqawasmi07@gmail.com (B.Q.); abdelrahmanalawaisheh@gmail.com (A.A.); 2Independent Researcher, Jerash 26126, Jordan; saifmedyu@outlook.com

**Keywords:** IVF, follicular fluid, biomarkers, metabolites, proteins, hormones, vitamins, microRNA

## Abstract

(1) Background: The follicular fluid (FF) comprises a large portion of ovarian follicles, and serves as both a communication and growth medium for oocytes, and thus should be representative of the metabolomic status of the follicle. This review aims to explore FF biomarkers as well as their effects on fertilization, oocyte, and embryo development, and later on implantation and maintenance of pregnancy. (2) Methods: This review was registered in the PROSPERO database with the ID: CRD42025633101. We parsed PubMed, Scopus, and Google Scholar for research on the effects of different FF biomarkers on IVF/ICSI outcomes in normo-ovulatory women. Included studies were assessed for risk of bias using the NOS scale. Data were extracted and tabulated by two independent researchers. (3) Results: 22 included articles, with a sample size range of 31 to 414 and a median of 60 participants, contained 61 biomarkers, including proteins, growth factors, steroid and polypeptide hormones, inflammation and oxidative stress markers, amino acids, vitamins, lipids of different types, and miRNAs. Most of the biomarkers studied had significant effects on IVF/ICSI outcomes, and seem to have roles in various cellular pathways responsible for oocyte and embryo growth, implantation, placental formation, and maintenance of pregnancy. The FF metabolome also seems to be interconnected, with its various components influencing the levels and activities of each other through feedback loops. (4) Conclusions: FF biomarkers can be utilized for diagnostic and therapeutic purposes in IVF; however, further studies are required for choosing the most promising ones due to heterogeneity of results. Widespread adoption of LC-MS and miRNA microarrays can help quantify a representative FF metabolome, and we see great potential for in vitro supplementation (IVS) of some FF biomarkers in improving IVF/ICSI outcomes.

## 1. Introduction

Assisted reproductive techniques (ARTs) such as in vitro fertilization (IVF) and intracytoplasmic sperm injection (ICSI) have become more prevalent, more affordable, and more culturally acceptable worldwide over the years [1]. They quickly became the last resort for a lot of couples suffering from infertility to have biological children of their own [2]. Yet despite its prevalence, and the well-established medical and surgical techniques used in IVF, there are still no established biochemical or molecular methods to evaluate the chance of IVF/ICSI success outside of research, regardless of the precision involved in applying its techniques.

Prior research on various methods, such as granulosa cell or polar body biopsy, has been done to predict IVF/ICSI outcomes. Some authors studied the expression of genes in the granulosa cells or in the oocyte itself, looking for specific molecular markers of oocyte quality [3]. Polar body biopsy is also sometimes used to screen oocytes with chromosomal defects deriving from errors in meiotic divisions [4]. Yet those techniques tend to be somewhat complicated, expensive, time-consuming, and invasive, and can even negatively affect the outcome of IVF [5]. Without a large enough sample, they might not be representative of the metabolomic status of the follicle as a whole.

The follicular fluid (FF) surrounding oocytes provides a suitable microenvironment for oocyte maturation and cellular communication [5,6,7]. The molecular components that comprise this fluid are promising candidates as representative biomarkers of follicular development, due to their theorized roles of influencing follicular maturation. In addition to the ease by which the FF can be collected during IVF, new retrieval techniques incur relatively little damage to the oocyte [8].

Past reviews focused on the role of FF as a predictor of IVF/ICSI success either in animals or in women suffering from ovulatory disorders such as polycystic ovary syndrome (PCOS) or endometriosis [5,9,10,11]. The PCOS and endometriosis reviews showed that these pathological conditions are associated with changes in FF metabolomics compared with normo-ovulatory women, which means their results may not be applicable to women suffering from other causes of infertility [10,11], which is why this review focuses on original research articles involving FF analysis specifically in normo-ovulatory women.

Our review aims to establish the possible role of FF constituents in predicting IVF/ICSI outcomes in normo-ovulatory women, including oocyte count, oocyte quality, fertilization rate, embryo quality, implantation rate, clinical or chemical pregnancy rate, miscarriage rate, and live birth rate. We hope in this review to spark further research regarding noninvasive methods to evaluate the expected IVF/ICSI outcomes and possible interventions that could be applied to improve outcomes.

## 2. Materials and Methods

### 2.1. Protocol

This systematic review was conducted in accordance with the Preferred Reporting Items for Systematic reviews and Meta-Analyses (PRISMA) 2020 statement [12]. PRISMA 2020 checklists for the abstract and the manuscript are provided in Appendix A. The protocol was registered in the international database of prospectively registered systematic reviews (PROSPERO) with the ID: CRD42025633101.

### 2.2. Information Sources and Search Approach

Research articles were extracted from two databases (PubMed, Scopus) and the search engine Google Scholar up to 1 October 2024, by creating separate, yet similar, search phrases for each database/search engine using keywords, operators, and search parameters—provided in Appendix A. We included English-written articles published since 2010, discussing IVF in normo-ovulatory human subjects that reported on FF biomarkers such as proteins, growth factors, steroid and polypeptide hormones, inflammation and oxidative stress markers, amino acids, vitamins, lipids of different types, microRNAs (miRNAs), etc., and on IVF/ICSI outcomes such as oocyte count, oocyte quality, fertilization rate, embryo quality, implantation rate, clinical or chemical pregnancy rate, miscarriage rate, and live birth rate.

Excluded articles were any study that did not mention follicular fluid (FF) levels of aforementioned biomarkers or focused only on their serum levels, or performed granulosa cell, cumulus cell, or polar body biopsy, and articles that included PCOS or endometriosis as causes of infertility or did not explicitly claim their exclusion, as well as animal studies. An updated search query was also performed on 6 March 2025, but did not yield any additional studies that fulfilled the inclusion criteria.

Articles were filtered at multiple steps, starting by eliminating articles with irrelevant titles, then screening their abstracts for relevance, and lastly retrieving articles with an accessible full text and reviewing it for relevant results based on the aforementioned inclusion and exclusion criteria.

### 2.3. Quality Assessment and Data Retrieval

The Newcastle-Ottawa Scale (NOS) was used to assess the quality and risk of bias of included studies after filtering by two independent researchers. Data were then extracted from eligible articles by another two independent researchers and summarized in tables. Extracted information included: study design, population characteristics, measured follicular fluid biomarkers, IVF/ICSI outcomes, and the findings of each study, including statistical measures such as *p*-values. Conflicts in quality assessment and data extraction were resolved by a majority vote of all reviewers.

## 3. Results

### 3.1. Study Selection

The total number of studies was 427, distributed as follows: 122 articles from PubMed, 70 articles from Scopus, and 235 articles from Google Scholar. Since Google Scholar is a search engine and not a database like PubMed and Scopus, we opted to use the PRISMA 2020 flow diagram for databases, registers, and other sources to better illustrate this difference (Figure 1) [12]. The flow diagram shows the filtering steps, at the end of which 22 eligible articles were included, 8 of which were cohort studies and 16 were case-control studies. The filtering and record retrieval steps involved all reviewers. The sample sizes of included studies ranged from 31 to 414 participants, with a median of 60 participants. All 22 studies passed the risk of bias assessment using the NOS scale, with cohort studies scoring ≥ 7 points and case-control scoring ≥ 8 points out of 9. Detailed scoring of each study is provided in Appendix A. Table 1 demonstrates the biomarkers included, categorized into nine main types, including the number of studies discussing each type. More details on each category of biomarkers and protocols followed in each study in Appendix A. IVF/ICSI outcomes discussed in each article are outlined in Table 2 below, and can be divided into early (before starting transfer cycles) and late (after starting transfer cycles) outcomes.

### 3.2. Proteins

Histidine-rich glycoprotein (HRG) is a multidomain plasma protein of hepatic origin that plays a role in regulating multiple processes including the complement system, which includes the opsonin C4B protein [13]. Sun X. et al. found that increased HRG levels in FF and decreased C4B levels in FF were significantly correlated with higher implantation rates (*p* < 0.001 and =0.011, respectively) [14].

Cathepsin B is a protease enzyme found in lysosomes that is responsible for the degradation of intracellular and extracellular proteins [15]. Bastu et al. found that higher levels of cathepsin B were significantly positively correlated with higher numbers of retrieved oocytes, number of MII (metaphase II, mature) oocytes, MII oocyte rate, fertilization rate, and clinical pregnancy rate (*p* = 0.033, =0.037, =0.032, =0.042, and <0.001, respectively) [16].

Matrix metalloproteinases (MMPs) are zinc-dependent proteolytic enzymes that play roles in tissue remodeling and organization of the extracellular matrix (ECM) during folliculogenesis [17]. The studies by Atabakhsh et al. and Bilen et al. found no correlation between MMP-2 activity and oocyte count, MII oocyte count, or fertilization rate. However, Atabakhsh et al. found a positive correlation between MMP-2 activity and embryo quality (*p* = 0.014). The studies disagreed on the significance of the correlation between MMP-2 and oocyte quality (*p* = 0.021 and N.S.). Both studies also found a positive correlation between MMP-9 and oocyte quality (*p* = 0.014 and 0.01), but no correlation was found with oocyte count or MII oocyte count. Additionally, Atabakhsh et al. found no correlation between MMP-9 activity and embryo quality. The studies disagreed on the significance of the correlation between MMP-9 and fertilization rate (*p* = 0.02 and N.S.) [18,19].

Lactoferrin is an iron-binding globulin with numerous immune and anti-inflammatory functions, and is also produced by theca cells (TCs) to influence follicular maturation [20]. Mostafa et al. found no significant difference in FF lactoferrin concentrations between women with a positive chemical pregnancy and a negative chemical pregnancy. Additionally, there was no significant correlation between lactoferrin levels and the number of retrieved, mature or fertilized oocytes, or embryo quality [21].

**Table 1 biomolecules-15-00443-t001:** FF biomarker categories distribution across different studies (some studies may contain more than one constituent or constituent type).

Biomarker Category	No. of Studies	No. of Biomarkers	Biomarkers
Proteins	5 [14,16,18,19,21]	6	Complement protein C4B, Histidine-rich glycoproteins (HRGs), Cathepsin B, Matrix metalloproteinase 2 (MMP-2), Matrix metalloproteinase 9 (MMP-9), and Lactoferrin
Growth factors	4 [22,23,24,25]	5	R-spondin2, Amphiregulin (AREG), Bone morphogenic protein-15 (BMP-15), Insulin-like growth factor-1 (IGF-1), and Stem cell factor (SCF)
Steroid hormones	6 [26,27,28,29,30,31]	6	Progesterone, Testosterone, Estradiol (E2), Dehydroepiandrosterone sulfate (DHEAS), 25-hydroxycholesterol (25-HC), and Deoxycorticosterone (DOC)
Polypeptide hormones	3 [16,29,32]	3	Anti-mullerian hormone (AMH), Relaxin, and Leptin
Inflammation and Oxidative stress markers	6 [33,34,35,36]	8	Interleukin-6 (IL-6), Monocyte chemotactic protein-1 (MCP-1), Coenzyme Q10 (CoQ10), Total antioxidant capacity (TAC), Nicotine glucuronide, 4,5-dihydroorotic acid (4,5-DHOA), 5,6-dihydrouridine (5,6-DHU), and Homocysteine (Hcy)
Amino acids and related metabolites	2 [30,31]	5	Phenylalanine, Leucine, Tryptophan, Maleylacetoacetic acid (MAAA), and Rhazidigenine Nb-oxide
Vitamins and related metabolites	4 [30,31,37,38]	7	Vitamin D, Vitamin A, Vitamin B6, Vitamin E, Lithocholic acid, 4-oxo-Retinoic acid (4-oxo-RA), and 13′-hydroxy-alpha-tocopherol (13′-HAT)
Lipids and related metabolites *	3 [28,30,31]	3 types, 17 factors	- 3 energy production lipids - 10 phospholipids - 4 fatty acids
MiRNAs	1 [28]	4	miR-26b-5p, miR-34a-5p, miR-145-5p, and miR-204-5p

* Detailed list in Appendix A.

**Table 2 biomolecules-15-00443-t002:** FF biomarker categories and discussed IVF/ICSI outcomes in each article (some studies contain multiple factors and may discuss different outcomes for each factor).

First Author	Biomarker(s) Category	O.C.	O.Q.	F.R.	E.Q.	I.R.	P.R.	M.R.	L.B.R.
Sun X. et al. [14]	Proteins					x			
Bastu et al. [16]	Proteins, Polypeptide hormones	x	x	x			x		
Atabakhsh et al. [18]	Proteins	x	x	x	x				
Bilen et al. [19]	Proteins	x	x	x					
Mostafa et al. [21]	Proteins	x	x	x	x		x		
Jasimkadim et al. [22]	Growth factors					x	x		
Wu et al. [23]	Growth factors					x	x		x
Mehta et al. [24]	Growth factors				x	x	x		
Celik et al. [25]	Growth factors						x		
Carpintero et al. [26]	Steroid hormones		x	x	x		x		
Yang et al. [27]	Steroid hormones, Inflammation, and Oxidative stress	x	x		x		x		
Habibi et al. [28]	Steroid hormones, Lipids, MiRNAs				x		x		
Alzubaidy et al. [29]	Steroid and Polypeptide hormones						x		
Song et al. [30]	Steroid hormones, Amino acids, Vitamins, Lipids							x	
Sun Z. et al. [31]	Steroid hormones, Inflammation and Oxidative stress markers, Amino acids, Vitamins, Lipids		x						
Ahmeid et al. [32]	Polypeptide hormones						x		
Buyuk et al. [33]	Inflammation and Oxidative stress						x		
Akarsu et al. [34]	Inflammation and Oxidative stress				x		x		
Abdul-Razzaq et al. [35]	Inflammation and Oxidative stress					x			
Ocal et al. [36]	Inflammation and Oxidative stress						x		
Ekapatria et al. [37]	Vitamins		x	x					
Gode et al. [38]	Vitamins				x		x		

Legend: O.C.—oocyte count, O.Q.—oocyte quality, F.R.—fertilization rate, E.Q.—embryo quality, I.R.—implantation rate, P.R.—pregnancy rate, M.R.—miscarriage or pregnancy loss rate, L.B.R.—live birth rate.

### 3.3. Growth Factors

R-spondin2 is a protein that belongs to the Wnt signaling pathway, and plays an important role in follicular growth and development [39]. Amphiregulin and bone morphogenic protein-15 (BMP-15) are growth factors belonging to the epidermal growth factor (EGF) and the transforming growth factor-β (TGF-β) families, respectively [40,41]. Jasimkadim et al. found a significant correlation between implantation rates and FF R-spondin2 and FF Amphiregulin levels (*p* = 0.015 and =0.008, respectively), but not between implantation rates and FF BMP-15 levels. Paradoxically, they found no significant differences in FF R-spondin2, FF Amphiregulin, or FF BMP-15 levels according to pregnancy rate [22]. In a similar study by Wu et al., higher FF BMP-15 levels were positively correlated with higher implantation rate, clinical pregnancy rate, chemical pregnancy rate, and live birth rate (*p* = 0.023, =0.016, =0.011, and =0.019, respectively) [23].

Insulin-like growth factor-1 (IGF-1) is a growth factor of hepatic origin, and can also be produced by granulosa cells (GCs). It has roles in follicle development and regulation of steroidogenesis by GCs [42,43]. Mehta et al. observed higher FF IGF-1 levels in women who achieved clinical pregnancy compared to women who did not (*p* = 0.0002). Embryo quality, implantation rates, and clinical pregnancy rates and were all higher in the high FF IGF-1 group (*p* < 0.0001, =0.0152, and =0.0272, respectively) [24].

Stem cell factor (SCF) is a paracrine and autocrine growth factor produced by GCs during follicular development to promote maturation and growth [44]. Celik et al. found no significant difference in FF SCF concentrations between women who achieved pregnancy compared to women who did not [25].

### 3.4. Steroid Hormones

A study by Carpintero et al. on the effects of steroid hormones on embryo quality and fertilization rate found that FF levels of progesterone, testosterone, estradiol (E2), and dehydroepiandrosterone sulfate (DHEAS) were higher in cases that achieved normal fertilization compared to cases of failed fertilization, but only progesterone levels were significant (*p* = 0.003). In terms of embryo quality, estradiol levels, estradiol/progesterone ratio and estradiol/testosterone ratio were all significantly higher in high-quality embryos compared to low-quality embryos (*p* = 0.01, <0.005 and =0.001, respectively) [26].

In the same study, E2 levels were significantly higher in women who achieved pregnancy compared to women who did not (*p* = 0.02), but other hormones’ levels were not [26]. On the other hand, Yang et al. found no significant differences in FF E2 levels between women who achieved pregnancy and women who did not [27].

As for progesterone, Habibi et al. and Alzubaidy et al. found its FF levels to be higher in women who achieved pregnancy compared to women who did not (*p* < 0.0001 and <0.001, respectively) [28,29].

In a study by Song et al. on recurrent pregnancy loss (RPL), DHEAS levels were higher in the group that suffered from a pregnancy loss compared to the control group (*p* = 0.0019), while 25-hydroxycholesterol (25-HC) levels were lower in the pregnancy loss group compared to the control group (*p* = 0.0069) [30].

Deoxycorticosterone (DOC) is a precursor in the mineralocorticoid synthesis pathway in the adrenal glands [45]. Sun Z et al. found that FF levels of DOC were negatively correlated with oocyte quality (*p* = 0.04) [31].

### 3.5. Polypeptide Hormones

Anti-mullerian hormone (AMH) is produced by GCs, and plays roles in oocyte development and selection of ovarian follicles prior to ovulation [46]. Alzubaidy et al. found that the FF levels of AMH were significantly higher in women who achieved chemical pregnancy compared to women who did not (*p* < 0.001) [29]. However, Bastu et al. found no significant correlation between FF AMH and clinical pregnancy rate, the number of retrieved oocytes, the number of MII oocytes, MII oocyte rate, or fertilization rate [16].

Relaxin is a polypeptide hormone produced by TCs that aids in ovulation [47]. Bastu et al. found no significant correlation between FF relaxin levels and pregnancy rate, the number of retrieved oocytes, the number of MII oocytes, MII oocyte rate, or fertilization rate [16].

Leptin is a polypeptide hormone mainly produced from adipocytes, but also from GCs and TCs. It aids follicular development and regulates steroidogenesis in GCs [48,49]. In the study by Ahmeid et al., FF leptin levels were not significantly different between women with positive chemical pregnancy and negative chemical pregnancy [32].

### 3.6. Inflammation and Oxidative Stress Markers

Interleukin 6 (IL-6) is a cytokine that is produced by immune cells and GCs. It plays an important role in ovulation [50]. Yang et al. found significantly higher levels of FF IL-6 in women who achieved pregnancy compared to women who did not (*p* = 0.016). FF IL-6 levels did not affect the number of retrieved oocytes or the number of mature oocytes. In terms of embryo development, higher FF IL-6 levels were significantly correlated with a lower embryo fragmentation grade (*p* < 0.0001), meaning a higher development potential for embryos, but not with blastomere cell count or symmetry [27].

The proinflammatory chemokine MCP-1 (monocyte chemotactic protein-1) also plays a role in ovulation induction [51]. Buyuk et al. tried to correlate between the FF levels of MCP-1 and clinical pregnancy rates, and found no significant correlation [33].

Coenzyme Q10 (CoQ10) plays an important role in the electron transport chain (ETC) and energy production, and is an essential antioxidant to prevent free radical formation [52]. Akarsu et al. correlated FF levels of CoQ10 to embryo quality and pregnancy rate. They found that CoQ10 levels were significantly higher in high-quality embryos than in low-quality embryos (*p* = 0.038). Additionally, FF CoQ10 was significantly higher in the group that achieved pregnancy compared to the group that did not (*p* = 0.044) [34].

Total antioxidant capacity (TAC) measures the ability of the FF to prevent follicular damage by free radicals [53]. Abdul-Razzaq et al. found significantly higher TAC levels in the FF of women who had successful implantation compared to those who suffered an implantation failure (*p* = 0.002) [35].

Nicotine is a toxic metabolite that accumulates in the body mainly from exposure to cigarette smoking, and has been shown to impact various functions in the reproductive system [54]. The compound 4,5-dihydroorotic acid (4,5-DHOA) is a precursor for the synthesis of uracil (U) nucleotides in RNA, and 5,6-dihydrouridine (5,6-DHU, D) is a modified nucleotide found in transfer RNA (tRNA) [55,56]. In the study by Sun Z et al., higher levels of nicotine glucuronide, 4,5-DHOA, and 5,6-DHU in the FF were significantly correlated with lower oocyte quality (*p* = 0.001, =0.007, and =0.005, respectively) [31].

Homocysteine (Hcy) is a non-essential non-proteinogenic amino acid that is part of the methionine cycle. High levels of Hcy can lead to the formation of free radicals and damage the vascular endothelium feeding follicles, ultimately leading to follicle damage and apoptosis [57,58]. Ocal et al. performed a subgroup analysis on couples with male factor infertility and found that FF Hcy levels were significantly higher in women who could not achieve pregnancy (*p* = 0.002) [36].

### 3.7. Amino Acids

In addition to their roles in proteinogenesis, amino acids can act as signaling molecules in various cellular pathways [59]. Song et al. found that higher FF levels of phenylalanine, leucine, and tryptophan were all significantly correlated with lower miscarriage rates (*p* = 0.0009, =0.0022, and =0.0003, respectively) [30]. On the other hand, Sun Z et al. found that Maleylacetoacetic acid (MAAA) and Rhazideginine Nb-oxide were negatively correlated with oocyte quality (*p* = 0.008 and =0.04, respectively) [31].

### 3.8. Vitamins

Ekapatria et al. found that high FF vitamin D levels had a significant positive correlation with oocyte quality (*p* = 0.01). Fertilization rates were also higher in the high vitamin D group, but the difference was not statistically significant (*p* = 0.13) [37]. The study by Song et al. on RPL found that vitamin D levels were lower in the group that suffered from a pregnancy loss compared to the control group (*p* = 0.0017) [30]. In the same study, FF levels of lithocholic acid, a bile acid that helps in the absorption of vitamin D [60], were also lower in the group that suffered from a pregnancy loss compared to the control group (*p* = 0.0008) [30].

Gode et al. correlated FF levels of vitamins A, B6, D, and E with embryo quality and pregnancy rate. Higher levels of FF vitamins A and B6 were significantly correlated with higher-quality embryos (*p* = 0.017 and 0.049, respectively). Levels of vitamins D and E were not significantly correlated with embryo quality (*p* = 0.999 and 0.505, respectively). There was no significant correlation between FF levels of all vitamins and clinical pregnancy rates (*p* = 0.830, =0.685, =0.971, and =0.673, respectively) [38].

The compound 4-oxo-Retinoic acid (4-oxo-RA) is a vitamin A breakdown product [61]. Sun Z et al. found that its FF levels were significantly correlated with lower oocyte quality (*p* = 0.03) [31], while Song et al. found FF levels of 13′-hydroxy-alpha-tocopherol (13′-HAT), an active metabolite of vitamin E, to be significantly lower in women who suffered from a pregnancy loss compared to a control group (*p* = 0.0019) [30].

### 3.9. Lipids

The different types of lipids included in our review and their related metabolites can be classified into three categories: lipids involved in energy production, membrane phospholipids, and fatty acids.

Triglycerides and diglycerides are broken into glycerol and fatty acids, which are transported into mitochondria by carnitine to enter β-oxidation for energy production [62]. In the study by Sun Z et al., Triglyceride (18:1/24:0/20:5) and Diglyceride (14:1/22:2) were significantly correlated with higher oocyte quality, while 3-hydroxynonanoyl-L-carnitine (3-HNC) was correlated with lower oocyte quality (*p* = 0.03, =0.04, and =0.04, respectively) [31].

Lysophosphatidylcholines (LysoPCs) and phytosphingosine form part of membrane phospholipids found in all cellular membranes [63]. LysoPCs are derived from phosphatidylcholine (PC) [64,65]. Sun Z et al. found that higher FF levels of LysoPC (14:0), LysoPC (16:0), LysoPC (18:0), and phytosphingosine were correlated with higher oocyte quality, while PC was correlated with lower oocyte quality (*p* = 0.02, =0.03, =0.04, =0.02, and =0.001, respectively) [31]. Conversely, Song et al. found that higher FF levels of LysoPC (16:0), LysoPC (18:0), LysoPC (18:1), LysoPC (18:2), LysoPC (20:3), LysoPC (20:4), and LysoPC (20:5) were correlated with higher miscarriage rates (*p* = 0.0073, =0.0007, =0.0068, =0.0091, =0.0012, =0.0009, and =0.0023, respectively) [30].

Docosahexaenoic acid (DHA), linoleate, and oleic acid are omega-3, -6, and -9 fatty acids, respectively [66,67]. Song et al. found that higher FF levels of DHA, linoleate, and oleic acid were associated with lower miscarriage rates (*p* = 0.0008, =0.0028, and =0.0052, respectively) [30].

Omega-6 fatty acids also include arachidonic acid (AA). Prostaglandin E2 (PGE2) is an AA metabolite with numerous reproductive functions. It regulates follicle development and mediates ovulation [68,69]. The study by Habibi et al. on women with a history of RIF found FF PGE2 levels to be higher in women who achieved pregnancy compared to women who did not, compared to a control group without a history of RIF (*p* < 0.0001) [28].

### 3.10. MiRNAs

In eukaryotic cells, microRNAs (miRNAs) play an important role in regulating gene expression [70]. Habibi et al. assessed the expression of several miRNAs and correlated their effects on embryo quality and pregnancy rates in women with a history of RIF. Among the tested miRNAs, they found that the upregulation of miR-26b-5p (*p* = 0.031) along with the downregulation of miR-34a-5p, miR-145-5p, and miR-204-5p (*p* = 0.048, =0.037, and =0.046, respectively) were more significant for the RIF group that achieved pregnancy compared to the group that did not. Moreover, they found the expression of miR-26b-5p to be higher in high-quality embryos (*p* = 0.004), while the expression of miR-34a-5p to be higher in low-quality embryos (*p* = 0.02). Expression levels of miR-145-5p and miR-204-5p were slightly higher in low-quality embryos, but the difference was not statistically significant (*p* = 0.8 and 0.2, respectively) [28].

The effects of the aforementioned FF biomarkers on IVF/ICSI outcomes are summarized in Table 3. The biomarkers that did not significantly correlate with any of the outcomes studied are excluded from the table. These include Ferritin, SCF, Testosterone, Relaxin, Leptin, MCP-1, and vitamin E.

## 4. Discussion

### 4.1. Follicular Fluid (FF)

The FF that lies in the follicular antrum and its components are derived from blood flowing through the network of capillaries surrounding theca cells (TCs), as well as from local synthesis by granulosa cells (GCs), TCs, and the cumulus–oocyte complex (COC) [71,72]. FF plays a crucial role in providing a supportive microenvironment for the development of oocytes. There is a clear correlation between specific biochemical characteristics of FF and the quality of oocytes, primarily driven by paracrine signaling between follicular cells [5,6,7]. Larger molecules require active transport, due to the blood-follicle barrier (BFB) at the level of the theca capillaries and the follicular basal lamina, which can also prevent the transport of large molecules secreted by GCs from leaving the cavity [71,73].

In this review, we aimed to explore the possible associations between different FF biomarkers on IVF/ICSI outcomes in normo-ovulatory women. Several components were found to be significantly associated with better outcomes while others were not.

### 4.2. Proteins

#### 4.2.1. Complement C4B

The immune system is active at the implantation site, especially dendritic cells, which seem to be important in decidual formation and blastocyst implantation [74]. A shift from Th1-dominant inflammatory responses to Th2-dominant responses is noticed in normal pregnancies carried to term [75], while an excessive response towards the Th1 response leads to adverse pregnancy outcomes, such as miscarriage, preeclampsia, and preterm birth [76].

The C4B complement protein participates in the classical and lectin complement immune pathways, which ultimately form the membrane attack complex (MAC complex) [77,78]. Disinhibition of the MAC complex at the maternal–fetal interface can lead to lysis of the embryonic trophoblasts, leading to miscarriage [79]. Mutations in genes encoding for complement regulators, such as CD46, CD55, and CD59, were previously linked to higher rates of recurrent pregnancy loss (RPL) [80]. These regulators normally prevent the overactivation of C4B and the MAC complex, thus preventing the aforementioned complications caused by the MAC complex [80,81].

#### 4.2.2. Histidine-Rich Glycoprotein (HRG)

HRG can be detected on the surface of several immune cells, such as macrophages and monocytes, and in alpha granules of platelets and megakaryocytes [82,83]. It can also be found on the surface of endothelial, endometrial, and myometrial cells of the female reproductive system, inside ovarian follicles and in embryos [84]. HRG interacts and binds with members of the complement components, such as the C1q activator of the classical complement pathway, and prevents their overactivation [13,85].

#### 4.2.3. Cathepsin B

Cathepsins are the most abundant lysosomal proteases. Cathepsin B can be found in GCs and the COC, and plays roles in follicle selection and rupture during ovulation [86,87]. Cathepsin B seems to play a role in steroidogenesis, where its downregulation results in decreased levels of CYP11A1 in TCs, the enzyme responsible for the conversion of cholesterol to pregnenolone [88]. It also seems to have roles in cell proliferation and remodeling of the ECM of endometrial cells by activating MMPs [89]. Through its role in ECM remodeling, Cathepsin B has been shown to cause the release of bound growth factors, inducing their effects on endometrial cells [90]. Growth factors seem to be plausible mediators for the effects of Cathepsin B and MMPs, especially with their positive effects on later IVF/ICSI outcomes. Although, in the studies included in our review, the correlation between most growth factors and earlier outcomes was not explored [22,23,24].

#### 4.2.4. Matrix Metalloproteinases (MMPs)

The expansion of the follicle requires remodeling of different components of the ECM surrounding the cells, changing their elasticity and preparing the follicle for ovulation [71]. MMPs are gelatinase enzymes involved in ECM remodeling, which is important for follicular growth and maintaining the structural integrity of the follicle [91,92].

During follicular growth, MMP-9 levels increase significantly with follicular diameter during folliculogenesis. While MMP-2 levels remained sufficient for its function, they also remained relatively stable throughout folliculogenesis [93,94]. The expression of both MMP-2 and MMP-9, as well as other related proteins, also seems to increase in blastocysts during embryogenesis [92]. In the studies by Atabakhsh et al. and Bilen et al., although the same outcomes were studied for each MMP, only MMP-9 seemed to be significantly correlated with fertilization rates, and only MMP-2 seemed to be significantly correlated with fertilization rates [18,19]. This may indicate a shift in MMPs’ expression and activity at different stages of the reproductive process, but further studies may be needed to confirm this shift.

### 4.3. Growth Factors

#### 4.3.1. R-Spondin2

R-spondin2 belongs to the R-spondin family of proteins, which plays a critical role in the Wnt signaling pathway [95]. The Wnt/β-catenin pathway was shown to induce the transcription of MMP-9, which in turn activates a heparin-binding EGF-like growth factor (HBEGF) that leads to increased progesterone production from GCs [96]. The Wnt/β-catenin pathway was also shown to increase endometrial receptivity, trophoblast cell proliferation, and angiogenesis at the maternal–fetal interface [97,98,99].

#### 4.3.2. Amphiregulin

Amphiregulin is part of the EGF family of proteins. Amphiregulin has been shown to play an important role in ovulation [100], as well as induce the expression of the StAR protein, necessary for progesterone production in GCs [101]. Amphiregulin has also been shown to increase the expression of both the vascular endothelial growth factor (VEGF) and vascular endothelial cadherin (VE-cadherin) [102,103], which induce angiogenesis essential for improving endometrial receptivity and implantation [104,105]. The expression of *AREG* (amphiregulin) mRNA also seems to increase in the endometrium during the window of implantation [106].

#### 4.3.3. Bone Morphogenic Protein-15 (BMP-15)

BMP-15 is expressed in oocytes during folliculogenesis alongside the growth differentiation factor 9 (GDF9) [107]. They play a role in follicle selection for ovulation, and their lack of expression can lead to an early block of folliculogenesis [108,109]. BMP-15 also influences the differentiation and proliferation of GCs, by increasing their sensitivity to FSH [110,111]. Cotreatment of COCs with BMP-15 and Amphiregulin shows they improve embryo quality in a manner that neither of them can achieve on its own [112].

BMPs play an important role in GCs differentiation, folliculogenesis, trophoblast differentiation, and maintenance of placental function [111,113,114,115], including BMP-4, which seems to inhibit the *CYP17A1* enzyme responsible for DHEAS synthesis, while leaving the rest of steroidogenesis intact [116].

#### 4.3.4. Insulin-like Growth Factor 1 (IGF-1)

IGF-1 is an autocrine and paracrine hormone primarily produced in the liver, stimulated by growth hormone (GH) [117]. Its action on GCs and later in embryonic cells starts by binding to the IGF-1 receptor (IGF-1R), which triggers intracellular signaling cascades stimulating cell proliferation and inhibiting the intrinsic and extrinsic apoptosis pathways [117,118]. IGF-1 seems to also play an important role in embryo implantation and later maintenance of the pregnancy. IGF-1 is an immune modulator, and seems to be able to suppress immune responses in the endometrium [119,120]. It also seems to increase the expression of the cyclooxygenase 2 (COX-2) enzyme in endometrial cells, increasing the production of PGE2 [121]. IGF-1 also seems to play a role in placental formation and facilitation of glucose, lipids, amino acids, and folate transport through its membrane to the fetus, ensuring optimal growth [122,123].

### 4.4. Steroid Hormones

#### 4.4.1. 25-Hydroxycholesterol (25-HC), Progesterone, and Estradiol (E2)

Cholesterol is the main precursor for steroidogenesis in ovarian follicles. TCs have low-density lipoprotein receptors (LDLRs) that allow them to uptake cholesterol from the serum [124]. The compound 25-HC is an oxysterol derived from cholesterol and can serve as a precursor for steroid hormone synthesis [125,126]. Progesterone receptors (PRs) and estrogen receptors (ERs) on the surface of GCs have been linked to follicular growth and maturation and prevention of apoptosis by altering the expression of several genes [127,128,129]. Decreased levels of ERs have also been linked to infertility and subfertility [130]. However, the effects of progesterone and E2 are not restricted to GCs, as they have been shown to have paracrine and autocrine effects through PR- and ER-activated signaling molecules that permeate extracellularly to TCs and the COC [129,131,132]. E2 also seems to increase the expression of Cathepsin B [133]. Paracrine effects of E2 have also been shown to prevent apoptosis in blastocysts and blastomeres, explaining its positive effects on embryo quality [134].

All types of PRs and ERs have multiple shared coactivators in both ovarian and endometrial tissues [127,128]. Along with 25-HC, they have been shown to modulate inflammatory responses in the endometrium, promoting maternal–fetal tolerance and ensuring optimal endometrial thickness, thus improving implantation rates [128,132,135,136]. They can also affect local immune cells as well as myometrial cells, preparing the uterus for better embryo receptivity [132,137]. They also help in the maintenance of early pregnancy by promoting vascularization of the trophoblast and later formation of the placenta, as well as prevention of its separation by inhibiting uterine contractions [138,139,140].

#### 4.4.2. Dehydroepiandrosterone Sulfate (DHEAS) and Deoxycorticosterone (DOC)

DHEAS has been shown to impair the function of dendritic cells [141], which are important for regulating the immune response and the establishment of a maternal–fetal interface [74]. DOC is a mineralocorticoid precursor for corticosterone, which is then converted into aldosterone [45]. The ovaries lack the enzymes necessary for glucocorticoid and mineralocorticoid synthesis, starting with the 21-hydroxylase and 11β-hydroxylase enzymes [45,124]. An error in these enzymes can impair corticosterone and aldosterone levels, while DOC accumulates [142]. The binding of mineralocorticoids to their receptors on GCs and TCs can have positive effects on oocyte development and maturation [143,144]. The elevated FF levels of DOC may indicate an enzymatic error that led to a decrease in corticosterone and aldosterone, thus negating their effects on oocyte quality.

### 4.5. Polypeptide Hormones

#### Anti-Mullerian Hormone (AMH)

AMH is a polypeptide hormone from the TGF-β superfamily produced by GCs in pre-antral follicles, and it plays an important role in the selection of follicles for ovulation [145,146]. High levels of AMH inhibit the binding of FSH to its receptor (FSHR) and downregulate the production of the aromatase enzyme that produces E2 in immature follicles, preventing any further maturation and selection for ovulation [147,148]. AMH can also inhibit the function of the 17α-hydroxylase and aromatase enzymes that are responsible for the production of progesterone and E2 [45].

Although the study by Alzubaidy et al. found it to significantly increase pregnancy rates [29], we did not find much evidence in past studies to support this claim. In the study by Bastu et al., AMH did not seem to be a good predictor for the IVF/ICSI outcomes studied, which included early and late outcomes [16]. Serum concentrations of AMH may be lower than needed to have a significant effect on the endometrium.

### 4.6. Inflammation and Oxidative Stress Markers

#### 4.6.1. Reactive Oxygen Species (ROS)

Normally, ROS can act as second messengers in multiple cellular signaling pathways important for mediating folliculogenesis, meiosis, ovulation, and embryonic development [149,150]. However, elevated ROS levels, exceeding the oxidative capacity of follicular cells and the FF, result in oxidative stress of the follicular environment, exerting damaging effects including meiotic arrest and apoptosis in oocytes, apoptosis of GCs, follicular atresia, or embryonic block [149,151,152]. Elevated ROS can initiate lipid peroxidation of membrane phospholipids, damaging mitochondrial and cellular membranes [153]. They also damage proteins, RNA, and even DNA. In the follicular environment, ROS are neutralized by an elaborate defense system that consists of enzymatic and non-enzymatic antioxidants [27,151].

#### 4.6.2. Interleukin-6 (IL-6)

ROS can induce inflammation by activating signaling pathways that lead to the production of pro-inflammatory cytokines, including IL-6. ROS activate multiple signaling pathways that upregulate the transcription of IL-6 and other cytokines [154,155]. Damage to cells incurred by ROS also leads to the formation of damage-associated molecular patterns (DAMPs), which further stimulate IL-6 production [156,157].

A wide spectrum of cells in the ovaries produce IL-6, including T and B lymphocytes, macrophages, and COCs. Syncytio- and extravillous trophoblasts also express IL-6 [158]. IL-6 can play a role in regulating embryonic implantation, endometrial decidualization, and establishing and maintaining maternal–fetal immune tolerance [159,160,161,162]. IL-6 also stimulates system A amino acid transporter activity in primary trophoblasts, increasing the uptake of essential amino acids [163].

IL-6 was also found to increase the expression of MMP-2 and MMP-9 in trophoblasts [164,165]. IL-6 secreted by trophoblasts also facilitates implantation by inducing regulatory T cells (Tregs) and suppressing cytotoxic T-cell activity, shifting the balance from a Th1 to a Th2 immune response and attenuating the local immune response of macrophages [166,167,168]. IL-6 can also control inflammation by minimizing the impact of other inflammatory cytokines such as IL-1β and TNF-α [157]. Expression of IL-6 and its receptor (IL-6R) has been shown to become upregulated during the implantation window, enhancing the adhesion of the embryo to the endometrial lining by promoting the expression of integrins, such as integrin β3, on endometrial cells [169]. IL-6 also stimulates the production of VEGF, aiding in angiogenesis and creating a supportive blood supply for the implanting of the embryo and maintenance of pregnancy [170,171].

Optimal FF levels of IL-6 are needed for it to exert its effects in a positive manner, which ensures optimal follicular maturation and improvement of implantation. FF IL-6 levels over 10 ng/mL were correlated with its pro-inflammatory effects, which disrupt the process of implantation. FF IL-6 levels below 3.67 ng/mL were shown to not be sufficient to enhance embryo adhesion or to exert its immune modulating effects on T cell responses, also resulting in implantation failure [50,172]. Amniotic fluid levels of IL-6 also seem to keep increasing throughout pregnancy until term. This increase may be from fetal or placental synthesis of IL-6, and may indicate an important role for IL-6 in the maintenance of pregnancy [173,174].

#### 4.6.3. Coenzyme Q10 (CoQ10) and Total Antioxidant Capacity (TAC)

CoQ10 has roles both in energy production and oxidative stress, by neutralizing ROS and inhibiting lipid peroxidation, which is critical for oocyte development and maturation [175,176]. Several studies have demonstrated that CoQ10 protects the ovarian reserve, counteracts physiological ovarian aging by restoring mitochondrial function, and increases the rate of embryo cleavage and blastocyst formation [177,178]. Any problem in the oxidative phosphorylation pathway can lead to the arrest of oocyte maturation, chromosomal misalignment, and compromised embryo development [177,179,180].

TAC is composed of the antioxidant capacity of total proteins, uric acid, bilirubin, carotenoids, tocopherol, and ascorbic acid [181]. Consumption of up to 50% of antioxidants in follicles and embryos takes place even before implantation, thus inadequate production of those antioxidants may lead to cellular damage before establishing a maternal–fetal interface capable of replenishing their levels [182,183,184]. Antioxidant supplementation after oocyte retrieval in IVF has been proposed for replenishing the TAC of the oocytes and embryos, and has shown great promise in improving IVF/ICSI outcomes [185,186].

#### 4.6.4. Nicotine Glucuronide

Nicotine glucuronide accounts for 3–5% of the converted nicotine in the human body before it is excreted in the urine [54], and is known to elevate ROS levels in follicles, inducing DNA damage and apoptosis [187]. Nicotine was also found to affect oocyte maturation and induce detrimental morphological effects on oocytes [188,189]. Although the study by Sun Z et al. did not mention smoking in the population characteristics, it could be the cause of the observed elevated nicotine glucuronide levels [31].

#### 4.6.5. Nucleic Acids

The compound 4,5-dihydroorotic acid (4,5-DHOA) is a precursor of the uracil (U) nucleotide that is essential for the synthesis of all RNAs in cells, and is converted into orotate by the action of the dihydroorotate dehydrogenase (DHODH) enzyme in mitochondria [55]. DHODH dysfunction can lead to decreased synthesis of uracil and accumulation of 4,5-DHOA. DHODH has been shown to have roles as an antioxidant for the preservation of mitochondrial membrane phospholipids, as well as mitochondrial energy production and ROS reduction [190,191]. DHODH dysfunction or inactivation was also shown to be correlated with increased rates of chromosome breaks, apoptosis, and ferroptosis [192,193].

The compound 5,6-dihydrouridine (5,6-DHU) is a modified pyrimidine base found in the tRNA, and plays an important role in its structural stability and its ability to interact with mRNA [56]. Disruptions in tRNA structure in follicular cells could impair protein synthesis, mitochondrial function, and overall cellular health, leading to reduced oocyte quality and follicular atresia. Studies have shown that modifications in tRNA can affect mitochondrial protein synthesis, which includes oxidative response proteins [194]. Oxidative stress has also been shown to increase the degradation of tRNAs and their modified nucleosides. These fragments may themselves trigger stress response pathways [195]. Thus, FF 5,6-DHU can be a viable marker for qualifying oxidative stress in follicles after retrieval.

#### 4.6.6. Homocysteine (Hcy)

Homocysteine is a non-essential non-proteinogenic amino acid produced in the metabolism of the essential amino acid methionine. A proportion of Hcy binds to serine to form cystathionine. This reaction is mediated by the enzyme cystathionine beta-synthase (CβS) and vitamin B6 [196]. Hyperhomocysteinemia (HHcy) may occur due to a deficiency of vitamins D, B6, or B9, as well as other vitamins. Vitamin D deficiency specifically has been shown to decrease CβS levels and increase Hcy leading to oxidative stress [197]. HHcy has been shown to induce damage in the vascular endothelium of the placenta and disrupt the coagulation cascade locally causing thrombosis, leading to complications such as preeclampsia, placental abruption, and RPL [58,198].

### 4.7. Amino Acids

#### 4.7.1. Phenylalanine, Leucine, and Tryptophan

In addition to their role in proteinogenesis as essential amino acids, phenylalanine, leucine, and tryptophan all seem to play important roles in the development of the trophoblast, its functions in implantation, and later on in supporting embryo development [199,200,201]. Leucine and tryptophan were previously shown to improve embryo development and decrease miscarriage rates [202,203]. Phenylalanine seems to increase the expression of the *LAT1* gene in trophoblasts, which plays roles in implantation, placental development, and placental transport of essential amino acids for fetal development [199,204]. Phenylalanine is also the main substrate for tyrosine synthesis. Both tyrosine and tryptophan are precursors for serotonin and melatonin synthesis, as well as other molecules [201,205,206]. Melatonin is a potent free radical scavenger; it neutralizes ROS and prevents oxidative stress [205]. Tyrosine is also essential for the formation of the benzoquinone structure in CoQ10 [207]. Tryptophan supplementation was also linked to higher oocyte counts and better oocyte quality through the action of melatonin on developing follicles [205]. Melatonin synthesized by trophoblasts has also been shown to play an important role in implantation and placental development [208]. Placental synthesized serotonin also plays an important role in fetal development and maintaining the stability of the maternal–fetal interface [209].

#### 4.7.2. Maleylacetoacetate (MAAA) and Rhazideginine Nb-Oxide

MAAA is an important intermediate in the catabolism of tyrosine, and is thus part of its degradation products. Elevation in FF levels of MAAA may indicate an increase in tyrosine degradation or a shift of tyrosine metabolism away from its active metabolites [210]. Rhazideginine Nb-oxide (the rhazidine base) may be related to tryptophan or may be one of its breakdown products, but the numerous chemical reaction steps to produce rhazideginine Nb-oxide from tryptophan may indicate a distant relationship between them [211,212]. It belongs to a class of vinca alkaloids, which are known to hinder cell proliferation [213,214].

### 4.8. Vitamins

#### 4.8.1. Vitamin D and Lithocholic Acid

Vitamin D seems to have a lot of connections with some of the other FF biomarkers we discussed earlier. In target cells, bioactive vitamin D binds to a specific nuclear receptor (VDR) to regulate the transcription of genes that are involved in a wide range of cellular processes [215]. VDR was found to be localized predominantly in pre-ovulatory follicles, in the oocyte, and GCs [216]. Vitamin D increases the transcription of *IGF-1*, its receptor (*IGF-1R*), the growth hormone receptor (*GHR*) in hepatic cells, which stimulates IGF-1 production, as well as the EGF family [217,218,219,220,221]. It also stimulates steroidogenesis by inducing the transcription of the *StAR* protein for cholesterol transport into TCs, *3β-HSD* responsible for progesterone synthesis, FSH receptor (*FSHR*), and *Aromatase* enzyme responsible for estradiol synthesis [222,223]. Progesterone, in turn, increases the transcription of *VDR*, and both vitamin D and progesterone seem to have the ability to bind to both of their respective receptors [224,225]. They both also seem to enhance Th2 and Treg immune responses, and stimulate the release of VEGF in endometrial cells [225]. Vitamin D also increases the transcription of *AMH* [226].

Vitamin D also has an immunomodulatory effect, improving maternal–fetal tolerance and decreasing rejection rates [227,228]. Vitamin D prevents IL-6 overactivation by interfering with STAT3 and NF-κB signaling [229,230,231]. It also enhances the expression of the *HOXA10* gene, which was shown to improve endometrial receptivity and implantation rates [232]. Vitamin D also enhances the transcription of multiple mitochondrial genes serving many functions, including energy production and response to oxidative stress and apoptosis prevention, thus contributing to the oxidative capacity of the follicle [218]. Vitamin D also enhances the expression of key antioxidant enzymes such as superoxide dismutase (*SOD*), catalase (*CAT*), glutathione reductase (*GR*), and glutathione peroxidase (*GPx*), which scavenge ROS and preserve TAC [233,234].

Lithocholic acid belongs to the family of bile acids, and is a vitamin D transporter [60]. Bile acids are transported from the serum to the ovarian follicle by both passive and active transport [235], the result of which is that their FF levels can reach twice their levels in the serum [236].

#### 4.8.2. Vitamin A and 4-Oxo-Retinoic Acid (4-oxo-RA)

Vitamin A (retinol) is a known potent antioxidant that protects the integrity of cell membranes [237], and along with its active metabolites (retinols), it has been shown to affect ovarian follicular growth, oocyte quality, steroidogenesis, and embryo quality [238]. It also modulates the expression of genes essential for oocyte growth and the expression of steroid hormone receptors [239].

The compound 4-oxo-RA is the degradation product of vitamin A, specifically, retinoic acid (RA) [61]. RA is first irreversibly converted into 4-hydroxy-retinoic acid (4-OH-RA), then into 4-oxo-RA [240]. Elevation of 4-oxo-RA in the FF may indicate an elevated activity of an underlying process that consumes vitamin A, such as high levels of oxidative stress.

#### 4.8.3. 13′-Hydroxy-Alpha-Tocopherol (13′-HAT)

The compound 13′-HAT is a vitamin E metabolite. It is a fat-soluble antioxidant, which can remove peroxyl radicals that result from the peroxidation of fatty acids, thus preventing the formation of ROS and reducing oxidative stress [241]. Studies indicate that elevated oxidative stress and elevated lipid peroxidation are related to the increased occurrence of spontaneous abortions [242].

#### 4.8.4. Vitamin B6

Vitamin B6 (pyridoxine) is a coenzyme mainly involved in the metabolism of amino acids, nucleic acids, and lipids, and plays a role in the glutathione antioxidant defense system alongside other B vitamins [243]. It may also react directly with peroxy radicals, scavenge radicals, and inhibit lipid peroxidation [244]. Studies have shown that the supplementation of vitamin B6 and other B vitamins can be beneficial to oocyte and embryo quality [245].

### 4.9. Lipids

#### 4.9.1. Energy Production Lipids

TGs and DGs are one of the most abundant lipids in oocytes, and are broken down into fatty acid chains that are transported to the mitochondria by acylcarnitines, such as 3-HNC, to be used for energy production through β-oxidation during oocyte maturation [246,247,248,249]. Accumulation of acylcarnitines in the cytoplasm of oocytes and embryos has been shown to increase peroxidation of these molecules instead of β-oxidation and generate ROS [250]. Thus, aberrations in energy production of TGs and DGs can hinder oocyte maturation and quality. DGs also activate a protein kinase C (PKC) responsible for regulating the cell cycle, cell survival, and apoptosis [251]. The use of a progestin-primed ovarian stimulation (PPOS) protocol by using medroxyprogesterone 17-acetate (MPA) has also been shown to increase the levels of both TGs and DGs in follicular fluid [252].

#### 4.9.2. Phospholipids

Phospholipids are the most abundant lipids in the cell membrane. They include lysophosphatidic acid (LPA), LysoPCs, Sphingosine-1 Phosphates (S1P), and Sphingophoryl Choline [63]. LysoPCs are derived by cleaving PC via the action of phospholipase A2 (PLA2) and/or by the transfer of fatty acids to free cholesterol via lecithin-cholesterol acyltransferase (LCAT) [64,65]. The binding of LysoPCs to their designated receptors activates multiple signaling pathways that are involved in oxidative stress and inflammatory responses [253,254].

LysoPCs have been shown to increase the release of pro-inflammatory cytokines, such as IL-6, IL-1β, TNF-α, and interferon-γ [255,256,257]. Furthermore, they increase the activation of B cells and macrophages [253,254,255], and enhance the expression of *Foxp3* in nTregs [258]. All of these effects can increase the inflammatory response at the maternal–fetal interface, interfering with the process of implantation.

LPA is produced by conversion of LysoPCs either by autotaxin (ATX) or PLA2 [65,259,260]. LPA has also been shown to increase the transcription of *FST* (follistatin) and *GDF-9* genes, which play important roles in folliculogenesis [261]. It also increases the transcription of *BCL2* and decreases the transcription of *BAX* genes, which prevents the apoptosis of follicular cells. In GCs, LPA increases the expression of the *FSHR* and *17β-HSD*, which stimulates the synthesis of E2 [262]. LysoPCs, LPA, and DGs are also precursors for cardiolipin, a unique phospholipid specific to mitochondrial membranes, which is important for oxidative phosphorylation and maintaining the mitochondrial oxidative capacity [263,264].

Phytosphingosine-1-phosphate (P1P), the phosphorylated form of phytosphingosine, can bind to S1P receptors [265,266]. S1P signaling activates the PI3K/Akt and MAPK/ERK pathways, which are commonly implicated in cell proliferation, differentiation, and apoptosis prevention [267]. The synergistic effect of EGFs with P1P on these pathways has been shown to greatly improve oocyte maturation compared to P1P alone [267,268].

P1P has also been shown to increase the expression of genes involved in cumulus expansion (*EGF* and *HAS2*), antioxidant enzymes (*SOD3* and *CAT*), and developmental competence (*OCT4*). It also influenced oocyte survival by shifting the ratio of BCL-2 to BAX while inactivating JNK signaling, which should improve the oxidative stress response in ovarian follicles [269]. Furthermore, EGF has been shown to increase *HAS2* expression, which encodes for the hyaluronan protein, which is an antioxidant located in the ECM of the COC responsible for scavenging free radicals [270].

PC is the main phospholipid in cellular membranes, endoplasmic reticulum, and Golgi apparatus [64,65]. It is also the main source of arachidonic acid (AA) synthesis, the precursor of PGE2 [271]. PC accumulation has been shown to increase the concentration of AA beyond normal levels, possibly shifting its metabolism in GCs to the lipoxygenase pathway, producing leukotrienes [272]. The pro-inflammatory properties of leukotrienes may be detrimental to oocyte quality [273].

#### 4.9.3. Fatty Acids

The essential fatty acids α-linoleic acid, linoleic acid, and oleic acid are the first substrates in the synthesis pathways of omega-3, omega-6, and omega-9 fatty acids, respectively [66,67]. Eicosapentaenoic acid (EPA) and docosahexaenoic acid (DHA) are part of the omega-3 synthesis chain started by α-linoleic acid, while arachidonic acid (AA) is part of the omega-6 chain of metabolites started by linoleic acid. AA and EPA are essential for the synthesis of series 2 and 3 eicosanoids, respectively, which include prostaglandins and leukotrienes [274,275]. Interestingly, PGE3 produced from EPA seems to antagonize the action of PGE2 on its receptors [275,276]. DHA activates ERK 1/2 signaling, which is known to be activated in cumulus cells to induce oocyte maturation [277,278]. DHA also seems to play a role in placental transport of long-chain fatty acids [279]. Oleic acid also seems to enhance amino acid transport in trophoblast and placenta cells, and increase live birth rates after embryo transfer in IVF [280]. Those fatty acids exert their actions through the PPAR signaling pathway [281]. PPAR signaling, especially PPARγ, was shown to be essential for placental development and pregnancy maintenance [282,283]. Supplementation of AA, EPA, and DHA has been shown to decrease the occurrence of neural tube defects (NTDs) in embryos [279,284]. On the other hand, the accumulation of linoleic acid and oleic acid seems to impair steroidogenesis and induce apoptosis in GCs, but these effects may be avoided by direct supplementation of AA and DHA [285,286].

PGE2 is the major prostaglandin synthesized in the ovaries and endometrium, and is a paracrine mediator of ovulation. Its FF levels reach their peak in the hours just before ovulation [287,288]. PGE2 was shown to induce the synthesis of members of the EGF family, including Amphiregulin and Epiregulin in GCs and the COC [289].

PGE2 seems to stimulate chemokines for trophoblast adhesion to the maternal decidua [290]. The synergistic effects of EGFs with PGE2 released from blastocysts enhances the expression of CXCR4 in endometrial cells, which is a receptor for the blastocyst chemokine CXCL12 on stromal cells of the endometrium [291,292,293]. The pathways activated by EGFs and PGE2 were also shown to mediate early embryo implantation by inducing the expression of genes involved in cell growth, differentiation, uterine receptivity, decidualization, and pregnancy maintenance [294,295]. PGE2 also seems to have a synergistic effect with progesterone on the activation of IL-11 in endometrial cells, which activates a protein kinase A (PKA) pathway involved in decidualization [296].

### 4.10. MiRNAs

MicroRNAs (miRNAs) are short single-stranded ribonucleotides with a length of about 22 nucleotides that play a role in regulating gene expression [297]. Their single-stranded structure allows them to bind to the 3′-UTR (3′ untranslated region) or 5′-UTR of mRNAs, inhibiting their translation into polypeptide chains that later produce functional proteins [298,299]. MiRNAs in the FF are mainly produced by GCs and transported by exosomes and microvesicles, which ensure the stability of miRNAs in this external environment and prevent their degradation while being transported into their designated targets, be it the oocyte or other GCs [300,301]. TCs and the COC also contribute to exosomal miRNAs in the FF [302].

The study by Habibi et al. investigated the expression of four genes targeted by the aforementioned miRNAs. The discussed target genes were *PTGS2*, *AREG*, *CAMK1D*, and *EFNB2*, although the targets for each miRNA were not specified [28]. We referred to multiple databases to confirm the previously mentioned targets and other expected targets for these miRNAs, namely, the miRDB sequence database, the NCBI RefSeq sequence database, and the Indian Statistical Institute TargetMiner mRNA database and database for putative microRNA-microRNA regulations (PmmR) [303,304,305,306,307,308,309,310,311].

#### 4.10.1. MiR-26b-5p and PTGS2

MiR-26b-5p targets the *PTGS2* gene across all databases [312,313,314], which encodes for the COX-2 enzyme that produces prostaglandins, including PGE2. Prostaglandins are also produced by the COX-1 enzyme encoded by the *PTGS1* gene [315]. Two isoforms of the *PTGS1* mRNA transcript also seem to be targets for miR-26b-5p, but not the gene itself [312,313,316,317]. Upregulation of miR-26b-5p has been shown to downregulate the expression of *PTGS2* [318].

The study by Habibi et al. showed higher levels of expression of miR-26b-5p, *PTGS2*, and PGE2 in the FF of women who achieved pregnancy [28], which should not be the case, since miR-26b-5p inhibits the expression of both COX genes, so its upregulation should decrease PGE2 levels and thus decrease pregnancy rates. These results may be explained by incomplete inhibition of both COX genes to a degree that still allows for PGE2 production, but this alone does not explain the elevated PGE2 levels [319]. Another insight arises when looking at expected targets of miR-26b-5p. One of these targets is the *HPGD* gene [312,320], which encodes the 15-PGDH enzyme responsible for PGE2 degradation [321]. Upregulation of miR-26b-5p may decrease the activity of 15-PGDH, allowing for PGE2 build-up in the FF.

#### 4.10.2. MiR-34a-5p and AREG

MiR-34a-5p targets the *AREG* (amphiregulin) gene according to miRDB, and although TargetMiner does not show it targeting its mRNA transcript, RefSeq shows a matching complementary sequence in the 3′-UTR of the transcript [322,323,324]. The study by Habibi et al. found the expression of *AREG* to be higher in women who achieved pregnancy [28]. This explains why upregulation of miR-34a-5p is associated with lower pregnancy rates, because it inhibits *AREG* expression and its associated positive effects.

#### 4.10.3. MiR-145-5p, MiR-204-5p, CAMK1D, and EFNB2

The last two miRNAs seem to both target two related genes: *CAMK1D* and *EFNB2* (ephrin-B2) [325,326]. MiR-34a-5p also seems to target certain isoforms of their mRNA transcripts [323,327,328]. CAMK1D is a serine/threonine kinase associated with several enzymes essential for steroidogenesis [329]. Specifically, CYP11A1 and 3β-HSD, which convert cholesterol to pregnenolone, and then to progesterone [330,331]. The study by Habibi et al. found the expression of *CAMK1D* to be higher in women who achieved pregnancy [28]. As before this explains the negative correlation between miR-34a-5p, miR-145-5p, and miR-204-5p and pregnancy rates.

Expression levels of *CAMK1D* and *EFNB2* have been shown to be linked [329], and ephrin-B2 has been shown to play an important role in angiogenesis, which is essential for embryo development [332]. The study by Habibi et al. shows the expression of *EFNB2* to be negatively correlated with embryo quality and pregnancy rates [28], which seems to contradict the results of its targeting miRNAs, as well as past research that correlated it with higher pregnancy rates [329].

#### 4.10.4. MiRNA-MiRNA Regulation

The PmmR database shows that these four miRNAs may play a role in regulating the expression of each other’s stem-loops. Each stem-loop is later split into active 5p and 3p miRNAs that share the same name as the stem-loop. We found that miR-145 (the stem-loop for miR-145-5p and miR-145-3p) regulates the expression of stem-loops miR-26a, miR-34b, and miR-204. Furthermore, both miR-26a and miR-34b regulate the expression of the miR-204 stem-loop [310,311]. However, the database does not show whether each miRNA upregulates or downregulates the expression of its target stem-loop. Currently, this is just speculation, but it may explain the expression levels of the discussed miRNAs in the group that achieved pregnancy compared to the group that did not. We also see potential in further research on these miRNA–miRNA regulation networks, as they may link the regulation of multiple seemingly unrelated metabolic pathways inside cells.

#### 4.10.5. Expected Targets

Careful reading through the miRDB database reveals some expected targets of the discussed miRNAs that are related to FF biomarkers mentioned in our review that had a significant effect on IVF/ICSI outcomes [312,322,325,326]. Please do note that some of these expected targets are not the exact molecules discussed in our review, but belong to the same family or cellular pathway, and not every miRNA had the same expected targets. We believe further research is still needed to confirm the relation between the expression of the aforementioned miRNAs and these expected targets. We shine a light on these expected targets in hopes of guiding future research on their significance to IVF/ICSI outcomes.

Some of the expected targets include complement proteins (not including C4B), members of the MMP family, Cathepsin V a cysteine protease similar to Cathepsin B, Wnt signaling pathway molecules including multiple R-spondins, Epiregulin an EGF similar to Amphiregulin, BMPs and their receptors, other members of the EGF and TGF-β families and their receptors, IGF-binding proteins and receptors (but not IGFs themselves), cholesterol-hydrolyzing enzymes, progesterone and estrogen receptors, receptors for interleukins including IL-6, oxidative stress response proteins, methionine/homocysteine cycle enzymes, and several lipid metabolism enzymes.

As we can see from Table 3, the IVF/ICSI outcomes linked to miRNAs as well as the other FF biomarkers related to them by target genes lie mostly in the second/latter half of the IVF cycle, such as implantation and pregnancy rate. Past studies have shown that some miRNAs are upregulated in the endometrium of women with a history of RIF, including miR-145-5p and miR-204-5p, as well as miR-34a-3p, which is cleaved from the same stem-loop as miR-34a-5p [333]. This may show their relation to implantation and maintenance of pregnancy not only in embryos, but also in the receiving endometrium. Still, we encourage further research into earlier IVF/ICSI outcomes, since miRNAs are produced locally in follicles even before retrieval, and may affect the development of oocytes.

### 4.11. Future Applications of Follicular Fluid Metabolomics

#### 4.11.1. Quantifying a Follicular Fluid Metabolome

At this point, we believe we established a representative metabolome of the FF that could potentially offer a way to determine the chances of IVF/ICSI success. Still, a diagnostic method that encapsulates this variable metabolome is needed to estimate those chances with high accuracy. Different methods for the detection of FF biomarkers were used by different studies (outlined in Table 3). These include ELISA (enzyme-linked immunosorbent assay), CLIA (chemiluminescent immunoassay), LC-MS (liquid chromatography with mass spectrometry), gelatin zymography, Western blot, latex nephelometry, peroxidase spectrophotometric assay, and the polymerase chain reaction (PCR). The use of each method will depend on its cost and availability in different regions. These methods can be used in parallel, given a sufficient FF sample is acquired. We aim to guide future research on simplifying and unifying the workflow needed to analyze FF samples both in research and hopefully in IVF centers.

ELISA was the most used method, and was used in 11 studies to quantify 14 of the significant biomarkers in our review, most of which are proteins in different categories. It was also used in the study by Habibi et al. to quantify PGE2, which is a fatty acid metabolite [28]. ELISA can also be used for the detection of other types of metabolites given that specific antibodies are developed for them, including hormones and vitamins [334,335]. However, the accuracy of ELISA compared to LC-MS in the detection of certain biomarkers is still a matter of debate [335,336,337]. ELISA microarrays can also allow for the simultaneous detection of up to 24 proteins in a single sample [338,339]. However, since ELISA is mainly designed for the detection of proteins, the use of specialized kits needed for the detection of other molecules may require the division of the FF sample across multiple assays, in addition to the cost or availability of each individual kit, we also could not find any sources that used ELISA microarrays for the detection of different types of metabolites simultaneously.

CLIA was also used in the studies by Carpintero et al. and Ekapatria et al. for the detection of steroid hormones and vitamin D, respectively [26,37]. CLIA still suffers from some of the shortcomings of ELISA in terms of sample division if a large number of molecules need to be tested, but CL-based microarrays (CL-MIA) can be promising for the detection of different types of metabolites with very little sample preparation [340]. Although, some studies have shown that CLIA may not be as accurate as LC-MS in the quantification of those molecules in some cases [341,342,343,344].

LC-MS, used in the studies by Song et al. and Sun Z et al., stands out for its ability to detect different types of metabolites in parallel, as well as a large number of metabolites per sample, including proteins, hormones, amino acids, vitamins, lipids, and other metabolites [30,31]. Other studies also established an LC-MS workflow that can be used for the detection of multiple types of metabolites in a single sample, which overcomes the sample division issue we discussed with ELISA [345]. Although LC-MS can be costly compared to ELISA for the quantification of a single metabolite or even a few metabolites, the cost can be brought down in the analysis of a large number of metabolites, especially with newer LC-MS workflows that bring down its cost even further without compromising on accuracy [346,347,348].

Nucleic acids, such as miRNA, have the advantage over other molecules in that their quantity can be multiplied after extraction from cells or sera using PCR. RT-qPCR (quantitative real-time PCR) and miRNA microarray chips are the most used and documented methods in research on miRNAs in the FF [28,301,349,350]. While RT-qPCR can be performed on its own, microarrays combined with multiplex PCR show great promise in cost-effectiveness and the time required for processing per sample [351,352], and allow for a more automated processing of multiple samples, up to 16 or even 24 at a time [352,353].

Some of the other methods mentioned are specific to certain biomarkers, and can be difficult to substitute depending on availability, such as various antioxidant assays for the measurement of TAC [354,355], which the peroxidase assay used in the study by Abdul-Razzaq et al. is only part of [35]. However, some studies show great promise in measuring TAC using LC-MS in low-volume samples such as FF without needing specialized kits, which may give a broader look into antioxidant activity in the FF [354,356,357]. Other methods can also be substituted for certain biomarkers, such as gelatin zymography, Western blot, latex nephelometry for MMPs, BMP-15, and homocysteine (Hcy) [18,23,36]. These biomarkers can be measured by ELISA, as shown by other studies included in our review and past research [19,22,358] or by LC-MS [359,360,361].

#### 4.11.2. An Interconnected Metabolome

We previously outlined some of the relationships that link several of the studied FF biomarkers together. These relationships include molecules involved in the same synthesis pathway, breakdown products, and mediating molecules that either activate or inhibit other cellular processes. Some of these relationships seem to form positive feedback loops that elevate the levels of certain factors concomitantly. We also discussed how aberrations in some synthesis pathways can lead to the accumulation of certain biomarkers resulting in a net negative effect on IVF/ICSI. Several of the discussed biomarkers influenced IVF/ICSI outcomes by either preventing apoptosis of oocytes and embryos, modulating immune system responses during implantation, promoting differentiation of trophoblasts, or enhancing placental transport of essential nutrients for optimal fetal growth. All of these relationships show that the FF metabolome is an interconnected one, with several molecules cooperating to exert specific effects. These relationships and effects are summarized and visualized in Figure 2 below.

#### 4.11.3. Therapeutic Applications

We previously discussed examples of how some FF biomarkers can be supplemented in the diet to improve their levels in the FF, such as vitamins, amino acids, and essential fatty acids [205,245,279,284]. We also discussed the ability to supplement antioxidants in oocytes and embryos after retrieval [185,186]. Some animal studies also showed that in vitro supplementation (IVS) of some amino acids improved oocyte and embryo quality [362,363,364]. Some studies found in vitro fatty acid supplementation during the thawing process to be beneficial for embryos [365,366,367]. We also discussed the effect of some FF biomarkers released by trophoblasts on the endometrium for the facilitation of embryo implantation, which suggests that their supplementation to embryos or the endometrium during implantation may be beneficial. Further research into the effect of IVS of various FF biomarkers may give the field of fertility medicine a new tool to improve IVF/ICSI outcomes, by correcting suboptimal levels of those biomarkers.

The study by Habibi et al. also showed that treating COCs taken from women with a history of RIF with FF taken from women without a history of RIF (considered normal FF in the study) significantly increased the expression of miR-26b-5p, miR-34a-5p, *PTGS2*, *AREG*, and *CAMK1D* in the COCs, while also significantly decreasing the expression of miR-145-5p, miR-204-5p, and *EFNB2* in them [28]. As discussed before, all of these effects (except for miR-34a-5p upregulation) should improve embryo quality and pregnancy rate. MiRNAs seem to be very promising candidates for both diagnostic and therapeutic applications in IVF, for both estimating chances of success and even improving IVF/ICSI outcomes in cases of suboptimal FF parameters. Their potential links to multiple pathways related to oocyte development and embryo implantation can make them the ultimate target for FF metabolomics in the future. While we do not see FF donors with a normal reproductive history being a feasible source of FF treatment outside of research, we believe there is potential for synthetic media that may mimic ideal FF constituents that could be used in such treatments, especially with the rise of technologies such as recombinant DNA (rDNA) and PCR [368,369], which could produce sufficient amounts of significant protein biomarkers and miRNAs for use in fertility clinics. We hope that these ideas do not remain just wishful thinking and inspire research and development in the pharmaceutical sector, as we believe such treatments could revolutionize the field of fertility medicine.

## 5. Limitations

We believe the results shown in our review are promising, since they encompass a fairly large number of biomarkers with mostly significant effects on various IVF/ICSI outcomes. As we have shown in the discussion, the effects of most biomarkers are consistent with their roles in various cellular pathways. Still, we believe more research is needed to confirm those results and expand on them, since most of the biomarkers were only explored by a single study or a few studies, and linked to a limited set of IVF/ICSI outcomes. This data heterogeneity also prevented us from performing a meta-analysis on our results. Although clinical trials were not excluded from our review, none of the eligible articles included clinical trials that explored the effect of FF biomarkers’ supplementation on IVF/ICSI outcomes, either maternally or in vitro. A more consistent and wider view on the effects of the included biomarkers in our review, as well as new ones, may aid in better understanding the mechanisms by which the FF metabolome affects IVF/ICSI outcomes, and allow for a more informed selection of ideal diagnostic and therapeutic targets.

## 6. Conclusions

In this review, we were able to establish a representative and wide FF metabolome and outline its effects on cellular pathways necessary for the development and differentiation of oocytes and embryos, as well as the various interactions that occur between its constituting biomarkers both inside and outside follicular cells. We provided a comprehensive view on the effects of FF biomarkers not only on IVF/ICSI outcomes shortly after oocyte retrieval, but also further down the line of the IVF process, up to reaching a full-term pregnancy. We compared the utility of different quantification methods of FF biomarkers and their ability to provide an accurate representation of the metabolomic status of the follicle. LC-MS stands out for its ability to quantify a fairly large number of biomarkers across different categories, and the potential link of miRNAs to various biomarkers and cellular pathways can make PCR and microarrays useful tools in the future, but some time is needed for their widespread adoption. We also discussed the potential of in vitro supplementation (IVS) of various FF biomarkers in improving IVF/ICSI outcomes and the possibility of creating synthetic media that mimic ideal FF constituents and its potential use to improve suboptimal levels of various FF biomarkers. It is still too early to identify the most promising FF biomarkers in terms of clinical diagnostic and therapeutic utility out of the ones included in our review, due to the low number of studies on each biomarker. There is still room for future research to improve upon the FF metabolome presented in our review and explore its potential utilization in IVF centers.

## Figures and Tables

**Figure 1 biomolecules-15-00443-f001:**
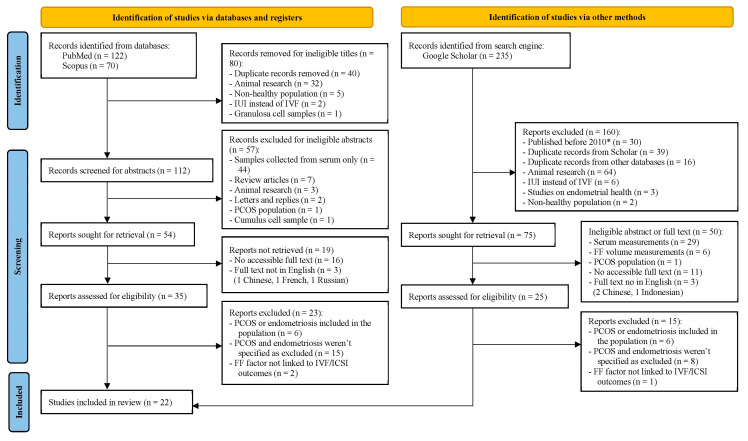
PRISMA 2020 flow diagram for new systematic reviews which included searches of databases, registers, and other sources. * Old articles appeared in the search query, although the publish year was specified to be 2010 or newer.

**Figure 2 biomolecules-15-00443-f002:**
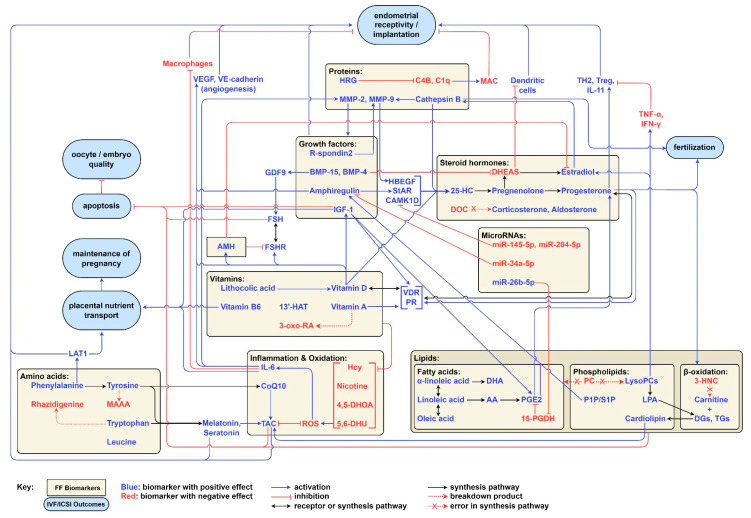
A visual summary of the relationships between FF biomarkers, mediating molecules, the immune system, and IVF/ICSI outcomes.

**Table 3 biomolecules-15-00443-t003:** Detection methods of FF biomarkers and their effects on IVF outcomes.

Category	Detection Method	Follicular Fluid Biomarker	#	O.C.	O.Q.	F.R.	E.Q.	I.R.	P.R.	M.R.	L.B.R.
Proteins	ELISA	Complement C4B [14]	1					↓			
		HRG [14]	1					↑			
		Cathepsin B [16]	1	↑	↑	↑			↑		
	ELISA and Zymography	MMP-2 [18,19]	2	N	↑/N	N	↑				
		MMP-9 [18,19]	2	N	↑	↑/N	N				
Growth factors	ELISA	R-spondin2 and AREG [22]	1					↑	N		
	ELISA and WB	BMP-15 [22,23]	2					↑/N	↑/N		↑
	ELISA	IGF-1 [24]	1				↑	↑	↑		
Steroid and Polypeptide hormones	ELISA and CLIA	Progesterone [26,28,29]	3			↑	N		↑/N		
		Estradiol (E2) [26,27]	2			N	↑		↑/N		
	LC-MS	DHEAS [30]	1			N	N		N	↑	
		25-HC [30]	1							↓	
		DOC [31]	1		↓						
	ELISA	AMH [16,29]	2	N	N	N			↑/N		
Inflammation and Oxidative stress markers		IL-6 [27]	1	N			↑		↑		
	LC-MS	Coenzyme Q10 [34]	1				↑		↑		
	Peroxidase assay	TAC [35]	1					↑			
	LC-MS	Nicotine, 4,5-DHOA, 5,6-DHU [31]	1		↓						
	Nephelometry	Homocysteine [36]	1						↓		
Amino acids and related metabolites	LC-MS	Phenylalanine, Leucine, Tryptophan [30]	1							↓	
		MAAA and Rhazidigenine [31]	1		↓						
Vitamins and related metabolites	LC-MS and CLIA	Vitamin D [30,37]	2		↑	N	N		N	↓	
	LC-MS	Lithocholic acid, 13′-HAT [30]	1						N	↓	
		Vitamin A and Vitamin B6 [38]	1				↑		N		
		4-oxo-Retinoic acid [31]	1		↓						
Lipids and related metabolites *		Energy metabolism lipids [31]	1		↑						
		Phospholipids [30,31]	2		↑					↑	
	LC-MS and ELISA	Fatty acids [28,30]	2						↑	↓	
MiRNAs	RT-qPCR	miR-26b-5p [28]	1				↑		↑		
		miR-34a-5p [28]	1				↓		↓		
		miR-145-5p and miR-204-5p [28]	1				N		↓		

Legend: O.C.—oocyte count, O.Q.—oocyte quality, F.R.—fertilization rate, E.Q.—embryo quality, I.R.—implantation rate, P.R.—pregnancy rate, M.R.—miscarriage or pregnancy loss rate, L.B.R.—live birth rate. * More details in Appendix A. # = number of studies discussing biomarker(s). ↑ = positive correlation, ↓ = negative correlation, N = non-significant, ↑/N = positive correlation in one study, but non-significant in another study. [empty cell] = biomarker was not studied in relation to the outcome.

## Data Availability

The original contributions presented in this study are included in the article/Appendix A. Further inquiries can be directed to the corresponding author.

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
