# Peer review of "Associations Between Follicular Fluid Biomarkers and IVF/ICSI Outcomes in Normo-Ovulatory Women—A Systematic Review"

_biomolecules, 2025, doi:10.3390/biom15030443_

Round 1
Reviewer 1 Report
Comments and Suggestions for Authors
The search for effective predictors of IVF success is of high relevance and practical importance, and follicular fluid in this context is a very promising object of research. It is obvious that the peer-reviewed review will be of interest to a large number of specialists, including due to the informative tables and the final scheme reflecting the possible relationships between follicular fluid components and IVF outcomes. I indeed liked the visual summary of the relationships between FF biomarkers (Figure 2), it is a really useful and clear summary. However, before MS can be accepted, some important corrections need to be made.
Major points
1. This review article is extremely verbose and needs to be significantly shortened to be clearer and more concise. In my opinion, there is no point in discriminating Discussion as a separate section in a review. The discussion actually starts in section 4.10.5. Up to this section, the text of Discussion largely repeats Results, but with other references. Here is just one example. Line 349 (Results): “phosphatidylcholine (PC), a major component of membrane phospholipids [65,66].” Line 1037 (Discussion): “PC is the main phospholipid in cellular membranes... [314].” Just about the same deals when other compounds are mentioned.
2. The reference list, containing 426 references, is also unjustifiably large, especially considering that the review is based on 22 papers selected by the authors for discussion. Such a large list of references significantly reduces the quality of the review and should be reduced to the necessary minimum.
In addition, the list of references should be carefully checked and brought into compliance with the rules of the journal. Not all references contain full imprint/periodical information and/or authors’ names (e.g., I.; … I.A.-… [18]; M.; … N.M.-O.J. of O. [21]; Wasiti, Dr. estabraq [106]. Conversely, there are some unnecessary information (e.g., ncbi.nlm.nih.gov [18]; scirp.org [21]; researchgate.net [35]; academia.edu [41], Springer [42], etc.; I did not note further).
3. Importantly, the bibliography (list of references) should contain references to the articles, not to the databases from which they were obtained. Below is the only example chosen at random, but there are many examples of this type of citation: Lines 2426–2427, Ref. [365]: Homo Sapiens Prostaglandin-Endoperoxide Synthase 2 (PTGS2), MRNA - Nucleotide - NCBI Available online: https://www.ncbi.nlm.nih.gov/nuccore/NM_000963.4 — actually it's about the article by Lv et al., appeared in Int. Endod. J. 57 (9), 1228-1246 (2024), on the basis of which the resource was created.
4. Some quotations in the text require correction. For example, when the authors talk about AMH in general (lines 245–247), they refer to [47], but I found this citation incorrect and rather strange, since the cited work deals with hens. The authors have excluded animal model studies from their paper (although some are still listed in the references) — so what relevance do even non-mammalian organisms like birds have to the aim of the review? In the given review, I also consider it not entirely appropriate to refer to the work carried out on zebrafish [120], which follows a rather general phrase: “BMP-15 is a growth factor of the transforming growth factor (TGF-β) superfamily of proteins” (lines 542–543). Line 938: How “spontaneous abortions” can be applied to fish? Ref. [278] talks about the morphological abnormalities of zebrafish embryos. Next, when leptin (lines 257-258) or cathepsin B (461–463) are mentioned, I don’t see points in referring to the work on cows [50,91], although cattle are still mammals — at least not birds or fish. There are more examples that could be given, but I’ll stop here.
5. The authors often, including in the title, talk about the effect of follicular fluid components on IVF/ICSI outcomes. Indeed, follicular fluid is the environment in which the oocyte develops. However, it cannot be stated with certainty that the level of all the compounds discussed directly affects the quality of the oocyte. In fact, the direct influence of a factor can only be proven experimentally, whereas the authors summarize data from clinical rather than experimental studies. I believe it would be more correct to talk about the relationship between the level of a particular compound and IVF/ICSI outcomes, at least in key parts of the review (title, summary, etc.).
Minor points
Reference numbers for papers by Song et al., Sun Z et al. (line 121) and Page MJ (line 168) are missed.
Table 1: The column “No. of studies” should be accompanied by the references.
Please correct the typo in line 5, right column: Inetrleukin must be Interleukin.
Lines 442–443: “multiple correlations for a single outcome indicate different results across studies” — this is not obvious from the table.
Vertical splits in all tables are hard to see.
Line 265: The word “role” is obviously missing after “an important”.
Table 3: I would swap the first three columns as follows: Сategory --> FF biomarker --> Method.
For “non significant” it is better to use the traditional abbreviation “n.s.” rather than “N”.
Line 674: Refs [166,167] are repeated twice.
Line 1420: I did not find Supplementary Materials (File S1: Search phrases) in the submitted items.
Author Response
Dear Reviewer,
We thank you and greatly appreciate your valuable comments, which we tried to address as much as possible and the following are the response for each of your comments.
Reviewer 1’s comments:
Comment 1: This review article is extremely verbose and needs to be significantly shortened to be clearer and more concise. In my opinion, there is no point in discriminating Discussion as a separate section in a review. The discussion actually starts in section 4.10.5. Up to this section, the text of Discussion largely repeats Results, but with other references. Here is just one example. Line 349 (Results): “phosphatidylcholine (PC), a major component of membrane phospholipids [65,66].” Line 1037 (Discussion): “PC is the main phospholipid in cellular membranes... [314].” Just about the same deals when other compounds are mentioned.
Response 1: We thank you for pointing this out. We removed the parts of the discussion that were repeated from the results, and tried to shorten the discussion as much as possible without compromising on the explanation of how each FF biomarker works in its respective pathway, or omitting any information we believe is necessary to the context of our review. The separation of the “Results” and “Discussion” sections is due to us following the PRISMA 2020 statement as well as the journal template in structuring our review. The results mostly reference the 22 included studies in our review, while the discussion mostly references additional articles used to discuss the structure and function of significant FF biomarkers, and the relationships between them and other molecules in their respective cellular pathways.
Comment 2: The reference list, containing 426 references, is also unjustifiably large, especially considering that the review is based on 22 papers selected by the authors for discussion. Such a large list of references significantly reduces the quality of the review and should be reduced to the necessary minimum.
Response 2: Based on your recommendation we tried to shorten the reference list as much as possible, and now our review contains 369 references. We agree that this is a large number of references compared to the number of included studies. Although our review only contains 22 included studies, these studies contain over 50 significant FF biomarkers, sometimes discussed in the context of multiple IVF/ICSI outcomes. We believe the references used are necessary to discuss each of the included FF biomarkers adequately and provide the full context of its theorized function in follicles, embryos and the endometrium. We know that our review is very ambitious in its approach, but we hope you understand our use of such a long list of references, even though it wasn’t our intention initially.
Comment 3: In addition, the list of references should be carefully checked and brought into compliance with the rules of the journal. Not all references contain full imprint/periodical information and/or authors’ names (e.g., I.; … I.A.-… [18]; M.; … N.M.-O.J. of O. [21]; Wasiti, Dr. estabraq [106]. Conversely, there are some unnecessary information (e.g., ncbi.nlm.nih.gov [18]; scirp.org [21]; researchgate.net [35]; academia.edu [41], Springer [42], etc.; I did not note further).
Response 3: We thank you for this comment. We were using a reference manager for importing and managing our references, which can encounter errors while importing references from some databases. Still we should’ve checked the references for any errors earlier. We manually edited these references to fix errors in author names or journals.
Comment 4: Importantly, the bibliography (list of references) should contain references to the articles, not to the databases from which they were obtained. Below is the only example chosen at random, but there are many examples of this type of citation: Lines 2426–2427, Ref. [365]: Homo Sapiens Prostaglandin-Endoperoxide Synthase 2 (PTGS2), MRNA - Nucleotide - NCBI Available online: https://www.ncbi.nlm.nih.gov/nuccore/NM_000963.4 — actually it's about the article by Lv et al., appeared in Int. Endod. J. 57 (9), 1228-1246 (2024), on the basis of which the resource was created.
Response 4: The use of the RefSeq database in our review was only for the comparison of complementary sequences between the discussed miRNAs and the mRNA transcripts of associated genes. Up to our knowledge RefSeq uses multiple sources to compile the sequences listed on their database, including multiple research articles and sequencing projects, and is updated based on new work on the genes. Unfortunately, we also couldn’t get access to the mentioned alternative study to confirm if it contains the full sequence of the mRNA transcript. We believe the use of genomic and transcriptomic databases is valuable to our study to explain the possible effects of miRNAs on the expression of their respective target genes, since they are a new area of research in the follicular fluid.
Comment 5: Some quotations in the text require correction. For example, when the authors talk about AMH in general (lines 245–247), they refer to [47], but I found this citation incorrect and rather strange, since the cited work deals with hens. The authors have excluded animal model studies from their paper (although some are still listed in the references) — so what relevance do even non-mammalian organisms like birds have to the aim of the review? In the given review, I also consider it not entirely appropriate to refer to the work carried out on zebrafish [120], which follows a rather general phrase: “BMP-15 is a growth factor of the transforming growth factor (TGF-β) superfamily of proteins” (lines 542–543). Line 938: How “spontaneous abortions” can be applied to fish? Ref. [278] talks about the morphological abnormalities of zebrafish embryos. Next, when leptin (lines 257-258) or cathepsin B (461–463) are mentioned, I don’t see points in referring to the work on cows [50,91], although cattle are still mammals — at least not birds or fish. There are more examples that could be given, but I’ll stop here.
Response 5: We thank you for this comment. The references of the studies on hens and zebrafish were removed and substituted for alternative studies in humans. We also tried as much as possible to substitute studies in other mammals with human studies based on your recommendation. Still we couldn’t find a suitable alternative for some of the studies in mammals, but we believe the use of these studies can be helpful to explain some of the effects of FF biomarkers, due to their similarity in structure and physiology between mammals and humans.
Comment 6: The authors often, including in the title, talk about the effect of follicular fluid components on IVF/ICSI outcomes. Indeed, follicular fluid is the environment in which the oocyte develops. However, it cannot be stated with certainty that the level of all the compounds discussed directly affects the quality of the oocyte. In fact, the direct influence of a factor can only be proven experimentally, whereas the authors summarize data from clinical rather than experimental studies. I believe it would be more correct to talk about the relationship between the level of a particular compound and IVF/ICSI outcomes, at least in key parts of the review (title, summary, etc.).
Response 6: We agree. The studies included in our review correlated the levels of different FF biomarkers to IVF/ICSI outcomes, and most of the biomarkers were significantly correlated with the outcomes, but we understand correlation doesn’t mean causation. The title of the review was changed to “Associations between Follicular Fluid Biomarkers and IVF/ICSI Outcomes in Normo-Ovulatory Women - A Systematic Review” to illustrate that this connection between biomarkers and outcomes is not a direct one. In the discussion we tried our best to explain the mechanism by which FF biomarkers can be linked to the outcomes through different cellular pathways and their effects on follicles, embryos and the endometrium. We also mentioned the lack of experimental studies/clinical trials in our included studies in the “Limitations” section.
Comment 7: Reference numbers for papers by Song et al., Sun Z et al. (line 121) and Page MJ (line 168) are missed.
Response 7: The first sentence was removed as part of our work in shortening the manuscript, since its content is shown in Table 1. But we agree that the references should’ve been added initially. The mention of the study by Page MJ et al. below Figure 1 was also removed. It was copied from the original file for the PRISMA 2020 diagram, but we don’t believe it should’ve been placed in the description of the figure, since the original paper was referenced in section 2.1 of material and methods.
Comment 8: Table 1: The column “No. of studies” should be accompanied by the references.
Response 8: References for the studies were added below their count.
Comment 9: Please correct the typo in line 5, right column: Inetrleukin must be Interleukin
Response 9: We thank you for pointing this out. It has been corrected.
Comment 10: Lines 442–443: “multiple correlations for a single outcome indicate different results across studies” — this is not obvious from the table.
Response 10: This refers to the fields in the table with “↑/N”, where in one study the factor was positively correlated with the outcome, while it wasn’t significant in the other study. This comment was removed from the table header, since we believe its explained more clearly in the footer of the table.
Comment 11: Line 265: The word “role” is obviously missing after “an important”.
Response 11: We thank you for pointing this out. We corrected this mistake in the manuscript.
Comment 12: Vertical splits in all tables are hard to see.
Response 12: Based on your recommendation we increased the weights of vertical splits in Tables 2 and 3.
Comment 13: Table 3: I would swap the first three columns as follows: Сategory --> FF biomarker --> Method.
Response 13: We thank you for this note. We swapped the table columns to be in the order Biomarker Catgeory à Detection method à FF biomarker(s). We believe this change made the results more clear, since the FF biomarkers became adjacent to both their respective results and detection methods.
Comment 14: For “non significant” it is better to use the traditional abbreviation “n.s.” rather than “N”.
Response 14: We agree that the use of the letter “N” in this context is non-conventional, but we were constrained by the space available in the journal’s template. The tables were initially meant to be in the landscape orientation. Although the conventional “N.S.” or the full form “non-significant” were used in the “Results” section. We also explained its use in the footer of Table 3.
Comment 15: Line 674: Refs [166,167] are repeated twice.
Response 15: We thank you for this note. We removed the repeated references.
Comment 16: Line 1420: I did not find Supplementary Materials (File S1: Search phrases) in the submitted items.
Response 16: We apologize for this inconvenience. We believe all supplementary files were provided with the manuscript. We will reupload another version of all supplementary files with the revised version of the manuscript.
Reviewer 2 Report
Comments and Suggestions for Authors
This is an interesting paper worthy of publication after major revision according to the following comments.
- Page 1, Title: The title should be better shorter, e.g. by deleting the words “Current Knowledge, Implications and Future Applications”.
- Page 1, Abstract: The Conclusion section should be extended and the Results section should be shortened and amended according to the comments presented below.
- Page 2, lines 85-88: Did the authors register their systematic review in a public registry such as PROSPERO (https://www.crd.york.ac.uk/prospero/) or the Open Science Framework (https://osf.io/) or Inplasy (https://inplasy.com/)? If not, they should better try to do this retrospectively or they should do this now and update their literature search.
- Page 2, line 88: The authors should provide a filled out PRISMA 2020 checklist (according to the authors’ reference 12: Page, M.J.; McKenzie, J.E.; Bossuyt, P.M.; Boutron, I.; Hoffmann, T.C.; Mulrow, C.D.; Shamseer, L.; Tetzlaff, J.M.; 1504 Akl, E.A.; Brennan, S.E.; et al. The PRISMA 2020 Statement: An Updated Guideline for Reporting Systematic 1505 Reviews. BMJ 2021, 372, doi:10.1136/BMJ.N71).
- Pages 2-3, lines 85-108, Materials and Methods: This section should be re-written, as it does not comply with the PRISMA statement; in particular, the authors should include each and every one of the items 5-15 of the PRISMA checklist.
- Page 3, lines 121-123: According to the PRISMA 2020 Statement, this information should be provided in the Materials and Methods, and the corresponding results in detail in the Results section, not in a supplementary file.
- Page 4, line 141: Please change “increased FF levels of HRG” to “increased HRG levels in FF”.
- Page 4, line 142: Please change “decreased FF levels of C4B” to “decreased C4B levels in FF”.
- Page 4, Flowchart: The two parts of the flowchart, i.e. data regarding PubMed and Scopus and data regarding Google Scholar should be better merged.
- Page 5, Table 1: The reference numbers of included studies should be added; e.g. in the first line “[14, 16, 18, 19, 21]” should be added either in the second column or in a new separate column.
- Results section: Item 20 of the PRISMA statement, i.e. “Results of syntheses” should be clearly reported. In particular, for each biomarker the number of studies evaluating this molecule should be reported. Furthermore, the authors should emphasize on clinically relevant information, especially on studies reporting information such as live-birth rate, miscarriage rate, pregnancy rate and implantation rates.
- Results section: Item 21 of the PRISMA statement, i.e. risk of bias should be reported.
- Pages 12-13: Table 3 should be moved to the Results section.
- Page 33: Figure 2 and subsection 4.11.2 should be moved to the Results section.
- Pages 11-33, Lines 396-1403: The Discussion section is unnecessarily long and it should be shortened considerably (e.g. to a maximum of 3-4 pages). Such a long text would me more appropriate for a thesis, rather than an article published in a journal. In particular, study characteristics have been adequately presented in the Results section in subsections 3.2.-3.9 (pages 3-11, lines 138-394). The authors should rather provide a concise “general interpretation of the results in the context of other evidence”, according to the PRISMA statement item 23a. The authors should especially discuss how many studies evaluating each biomarker have been conducted, as well as clinically relevant information, especially what is known regarding the live-birth rates, the miscarriage, the pregnancy and the implantation rates.
- Page 34, Conclusions section: The authors should add conclusions regarding clinically relevant outcomes and prospects for future research. Is there any biomarker or method used to analyze the follicular fluid worthy to be further investigated in the future? Does any biomarker appear to be more important than others?
Author Response
Dear Reviewer,
We thank you and greatly appreciate your valuable comments, which we tried to address as much as possible and the following are the response for each of your comments.
Reviewer 2’s comments:
Comment 1: Table 1, most categories have clear functions, for example, the markers are growth factors or hormones. But several categories are not, including proteins, Amino acids and related metabolites and Lipids and related metabolites*. Can they be put together as a new category?
Response 1: We tried as best as we could to categorize the included FF biomarkers according to their functions, but that wasn’t possible for all biomarkers, so we relied on their structure for categorization. For example in proteins, HRG has functions in multiple cellular processes, C4B is part of the innate immune system and the enzymes Cathepsin B and MMPs are involved in the breakdown of peptides and ECM remodeling. In the discussion we mentioned some of the ways in which these molecules can be related other than their structure. For example in amino acids, we found that some amino acids share secondary metabolites that may explain their effects, or that some other molecules are breakdown products of amino acids. We also subcategorized lipids into three categories based on their structure and metabolic pathways. In the end we found many connections between FF biomarkers in different categories, and illustrated these connections in Figure 2. We believe categorizing such a varied metabolome is difficult, and is not the main focus of our review, but we tried to simplify the categorization as much as possible, and making changes to it would lead to drastic changes in the structure of our review.
Comment 2: So many markers are list in this review. Can the authors just list the most promising markers in FF on IVF/ICSI in the end?
Response 2: We believe it’s still too early to specify certain biomarkers as being the most promising or the most important, due to the low number of studies on each biomarker. Especially since over 50 biomarkers seem to be significantly correlated to their respective outcomes in our results, and the various connections between those biomarkers illustrated in the discussion. We believe more research is still needed to confirm their correlation with IVF/ICSI outcomes. Still we added the two detection methods that we believe can be useful for future research in the “Conclusion” section, which were LC-MS and miRNA microarrays, but we still await confirmation on their utility in this field.
Reviewer 3 Report
Comments and Suggestions for Authors
The follicular fluid (FF) in follicles serves as both a communication and growth medium for oocytes, and thus should be representative of the metabolomic status of follicles. The authors aimed to review FF biomarkers as well as their effects on fertilization, oocyte and embryo development, and later on implantation and maintenance of pregnancy. They collected the related papers and found that most of the studied factors had significant effects on IVF/ICSI outcomes. Finally, they concluded that FF biomarkers can be utilized for diagnostic and therapeutic purposes in IVF. This is a valuable topic. Some concerns may be discussed for the improvement of review.
Table 1, most categories have clear functions, for example, the markers are growth factors or hormones. But several categories are not, including proteins, Amino acids and related metabolites and Lipids and related metabolites*. Can they be put together as a new category?
So many markers are list in this review. Can the authors just list the most promising markers in FF on IVF/ICSI in the end?
Author Response
Dear Reviewer,
We thank you and greatly appreciate your valuable comments, which we tried to address as much as possible and the following are the response for each of your comments.
to would to like to draw your attention as well that we had included the PRISMA checklists in the supplementary files
Reviewer 4’s comments:
Comment 1: Page 1, Title: The title should be better shorter, e.g. by deleting the words “Current Knowledge, Implications and Future Applications”.
Response 1: The title was shortened based on your recommendation to become “Associations between Follicular Fluid Biomarkers and IVF/ICSI Outcomes in Normo-Ovulatory Women - A Systematic Review”.
Comment 2: Page 1, Abstract: The Conclusion section should be extended and the Results section should be shortened and amended according to the comments presented below.
Response 2: The abstract was edited based on your recommendation.
Comment 3: Page 2, lines 85-88: Did the authors register their systematic review in a public registry such as PROSPERO (https://www.crd.york.ac.uk/prospero/) or the Open Science Framework (https://osf.io/) or Inplasy (https://inplasy.com/)? If not, they should better try to do this retrospectively or they should do this now and update their literature search.
Response 3: At the time of sending the first version of the manuscript, the PROSPERO registration process wasn’t complete. We since added the PROSPERO registration statement in section 2.1 of materials and methods.
Comment 4: Page 2, line 88: The authors should provide a filled out PRISMA 2020 checklist (according to the authors’ reference 12: Page, M.J.; McKenzie, J.E.; Bossuyt, P.M.; Boutron, I.; Hoffmann, T.C.; Mulrow, C.D.; Shamseer, L.; Tetzlaff, J.M.; 1504 Akl, E.A.; Brennan, S.E.; et al. The PRISMA 2020 Statement: An Updated Guideline for Reporting Systematic 1505 Reviews. BMJ 2021, 372, doi:10.1136/BMJ.N71).
Response 4: We sent a filled out PRISMA 2020 checklist alongside the second version of the manuscript with the revised title. We will also send an updated version with the revised version of the manuscript.
Comment 5: Pages 2-3, lines 85-108, Materials and Methods: This section should be re-written, as it does not comply with the PRISMA statement; in particular, the authors should include each and every one of the items 5-15 of the PRISMA checklist.
Response 5: We followed the PRISMA 2020 checklist ever since we started working on our review, and we believe we adhered to it as best as we could and filled out most of the items required to complete checklist, except for performing a confidence interval analysis, due to the low number of studies for each factor. The PRISMA 2020 paper states that “PRISMA 2020 is intended for use in systematic reviews that include synthesis (such as pairwise meta-analysis or other statistical synthesis methods) or do not include synthesis (for example, because only one eligible study is identified).”
Comment 6: Page 3, lines 121-123: According to the PRISMA 2020 Statement, this information should be provided in the Materials and Methods, and the corresponding results in detail in the Results section, not in a supplementary file.
Response 6: The search phrases we used on the two databases and search engine are over 500 words or 4000 characters long, and would’ve taken considerable space in the manuscript. Due to the open access nature of the journal they can still be accessible for all readers. And we believe this still adheres to the PRISMA 2020 checklist. The PRISMA 2020 paper states that “if the relevant information for some items already appears in a publicly accessible review protocol, referring to the protocol may suffice. Alternatively, placing detailed descriptions of the methods used or additional results (such as for less critical outcomes) in supplementary files is recommended.” It also doesn’t place constraints on the location of reporting each item, as the paper states “although PRISMA 2020 provides a template for where information might be located, the suggested location should not be seen as prescriptive; the guiding principle is to ensure the information is reported.”
Comment 7: Page 4, line 141: Please change “increased FF levels of HRG” to “increased HRG levels in FF”. Page 4, line 142: Please change “decreased FF levels of C4B” to “decreased C4B levels in FF”.
Response 7: These changes were made based on your recommendation.
Comment 8: Page 4, Flowchart: The two parts of the flowchart, i.e. data regarding PubMed and Scopus and data regarding Google Scholar should be better merged.
Response 8: The PRISMA 2020 statement website contains two main types of flow diagrams: one for databases and registers only, and one for databases, registers and other sources. We mentioned that Google scholar is a search engine, and thus we believe the second flow diagram is more suitable for our review. The flow diagram illustrated in the original PRISMA 2020 paper is also similar to the one used in our review.
Comment 9: Page 5, Table 1: The reference numbers of included studies should be added; e.g. in the first line “[14, 16, 18, 19, 21]” should be added either in the second column or in a new separate column.
Response 9: We agree. The reference numbers were added below the number of studies in each category.
Comment 10: Results section: Item 20 of the PRISMA statement, i.e. “Results of syntheses” should be clearly reported. In particular, for each biomarker the number of studies evaluating this molecule should be reported. Furthermore, the authors should emphasize on clinically relevant information, especially on studies reporting information such as live-birth rate, miscarriage rate, pregnancy rate and implantation rates.
Response 10: Based on your recommendation we added a column to Table 3 that lists the number of studies for each biomarker or group of biomarkers. Each biomarker is also accompanied by the reference of its original studies. We also believe we discussed the clinically relevant information regarding each biomarker in detail in the discussion, depending on the significance of the correlation between biomarkers and outcomes.
Comment 11: Results section: Item 21 of the PRISMA statement, i.e. risk of bias should be reported.
Response 11: This was an oversight by us. Although we performed the risk of bias assessment early in our review process, we didn’t initially include it because it didn’t exclude any studies. We added it in section 2.3 in materials and methods in the second version of the manuscript as well as the newly revised version.
Comment 12: Pages 12-13: Table 3 should be moved to the Results section.
Response 12: Table 3 was moved to the end of the Results based on your recommendation.
Comment 13: Page 33: Figure 2 and subsection 4.11.2 should be moved to the Results section.
Response 13: Subsection 4.11.2 and Figure 2 are based on the connections between FF biomarkers explored in the discussion, not on the results extracted from the included studies. We believe placing it earlier in the review may be confusing to readers, since these connections wouldn’t be mentioned before the figure.
Comment 14: Pages 11-33, Lines 396-1403: The Discussion section is unnecessarily long and it should be shortened considerably (e.g. to a maximum of 3-4 pages). Such a long text would me more appropriate for a thesis, rather than an article published in a journal. In particular, study characteristics have been adequately presented in the Results section in subsections 3.2.-3.9 (pages 3-11, lines 138-394). The authors should rather provide a concise “general interpretation of the results in the context of other evidence”, according to the PRISMA statement item 23a. The authors should especially discuss how many studies evaluating each biomarker have been conducted, as well as clinically relevant information, especially what is known regarding the live-birth rates, the miscarriage, the pregnancy and the implantation rates.
Response 14: We thank you for this comment. We have since shortened the discussion section considerably, ensuring that we don’t remove any information we believe is necessary to the context of our work. This is also caused by the large number of significant of FF biomarkers included in our review, for which we tried to explain the possible mechanisms by which they exert their effects and the various connections between them. We know that our review is very ambitious in its approach, but we believe the information included in the discussion is necessary to provide the full context of this review, and its ability to guide future research. Although it wasn’t our intention initially for the review to be this long.
Comment 15: Page 34, Conclusions section: The authors should add conclusions regarding clinically relevant outcomes and prospects for future research. Is there any biomarker or method used to analyze the follicular fluid worthy to be further investigated in the future? Does any biomarker appear to be more important than others?
Response 15: We thank you for this comment. Based on your recommendation we added the two detection methods that we believe can be useful for future research in the “Conclusion” section, which were LC-MS and miRNA microarrays. We believe it’s still too early to specify certain biomarkers as being the most promising or the most important, due to the low number of studies on each biomarker. Especially since over 50 biomarkers seem to be significantly correlated to their respective outcomes in our results, and the various connections between those biomarkers illustrated in the discussion.
Round 2
Reviewer 1 Report
Comments and Suggestions for Authors
I have carefully read the new version of the manuscript and found that the authors have done a great deal of work to improve it. They have responded to my comments and observations with detailed responses. In cases where the authors disagree with my opinion, their arguments seem well-founded, although I still do not agree with everything. For example, I do not consider the lack of access to the original sources a compelling argument. I still believe that the review could have been shortened even further. However, I appreciate the work done and believe that the article in its present form can be accepted for publication.
Author Response
I have carefully read the new version of the manuscript and found that the authors have done a great deal of work to improve it. They have responded to my comments and observations with detailed responses. In cases where the authors disagree with my opinion, their arguments seem well-founded, although I still do not agree with everything. For example, I do not consider the lack of access to the original sources a compelling argument. I still believe that the review could have been shortened even further. However, I appreciate the work done and believe that the article in its present form can be accepted for publication.
Response: Thank you for your comments, your former advice is greatly appreciated
Reviewer 2 Report
Comments and Suggestions for Authors
This paper still needs major revision according to the following comments.
- Abstract: The authors have provided a filled out PRISMA checklist for the Abstract and answered “yes” to items 1-10 and no to items 11 and 12. However, they did not mention the inclusion and exclusion criteria according to item 3, furthermore, they did not mention any number of participants according to item 7; in addition, the title of the fourth subsection is “Conclusions” instead of “Discussion”; finally, the answer to item 12 should have been “yes” instead of “no” and the PROSPERO registration ID should have been provided in the second subsection.
- Line 107: The authors should add that the PRISMA checklists are available in certain supplementary files.
-Line 112: The authors should update the literature search beyond “October 1, 2024”.
- Line 114: The name and number of the “supplementary file” should be provided, i.e. “in a supplementary file” should be changed to “in supplementary file X”.
- Lines 111-129: The authors have failed to comply with items 5-15 of the PRISMA checklist and the answers given in the last column of the PRISMA checklist (“Location where item is reported”) are inaccurate and do not correspond to the exact location In the text. For example, the authors state that the location in the text for item 5 of the PRISMA checklist is line 92; however, item 5 is a part of Methods, whereas line 92 is a part of the Introduction!
- Page 4, Flowchart: The two parts of the flowchart, i.e. data regarding PubMed and Scopus and data regarding Google Scholar should be better merged. Why did the authors classify Google Scholar under “other methods” rather than “databases and registers”?
- Results section: Item 21 of the PRISMA statement, i.e. risk of bias should be reported.
- Pages 11-25, Lines 599-3433: The Discussion section is now shorter, yet it remains unnecessarily long and it should be shortened considerably (e.g. to a maximum of 3-4 pages), as such a long text would me more appropriate for a thesis, rather than an article published in a journal. The authors should try to provide a concise “general interpretation of the results in the context of other evidence”, according to the PRISMA statement item 23a. Furthermore, the authors should focus on clinically relevant information.
Page 25, Conclusions section: Expressions such as “we believe” (line 3435 and 3451) and “we hope” (line 3438 and 3452) should be avoided. Authors should provide evidence and firm evidence-based conclusions rather than beliefs or hopes. The authors should better focus on clinically relevant outcomes. Is there any particular biomarker worthy to be further investigated in the future? Does any biomarker appear to be more important than others?
Author Response
Dear Reviewer,
we would like to thank you for your comments, kindly find our responses in the following:
Comment 1: Abstract: The authors have provided a filled out PRISMA checklist for the Abstract and answered “yes” to items 1-10 and no to items 11 and 12. However, they did not mention the inclusion and exclusion criteria according to item 3, furthermore, they did not mention any number of participants according to item 7; in addition, the title of the fourth subsection is “Conclusions” instead of “Discussion”; finally, the answer to item 12 should have been “yes” instead of “no” and the PROSPERO registration ID should have been provided in the second subsection.
Response 1: This is correct. Item 3 was answered “Yes” based on an earlier version of the abstract that included the exclusion criteria as well as other information, but was later shortened to fit the journal’s guidelines. We would’ve liked to include all the items listed the abstract checklist, but we were constrained by the journal’s limit on abstract length of up to 200 words, which we already exceeded. We since changed the answer to item 3 to “No” and added the registration information for item 12 based on your recommendation.
As for item 7, it would be impossible to list the number of participants from all studies in the abstract. Instead we included the number of FF biomarkers included in our review along with their categories. The following item 8 also uses the word “preferably” for the number of studies and participants, which indicates that they may not be mandatory. The use of “Conclusions” instead of “Discussion” was due to the journal’s template, and it also contains parts of the “Discussion”, “Limitations” and “Conclusion” sections from the manuscript. As we mentioned in our previous response, the PRISMA 2020 paper doesn’t place constraints on the location of reporting each item, as long as the item is reported.
Comment 2: Line 107: The authors should add that the PRISMA checklists are available in certain supplementary files.
Response 2: Based on your recommendation we added the PRISMA checklists as part of supplementary files.
Comment 3: Line 112: The authors should update the literature search beyond “October 1, 2024”.
Response 3: We checked the databases for any new articles using the same search phrases from our initial search strategy, and found 7 studies published after October 1st 2024 (3 from PubMed, 1 from Scopus and 3 from Google Scholar). However none of these studies were eligible to be included in our review. Reasons for exclusion include a study on animals, 3 studies where samples were only taken from serum, one study that included participants with PCOS and endometriosis, one study on endometrial scratching (out of scope for our review) and one study where the full-text was in Chinese, which prevented us from confirming if the population characteristics fulfill our review’s inclusion criteria.
The search results for Scopus and Google Scholar also showed no change in the number of retrieved articles (70 and 235, respectively). For Scopus we suspect this is due to a study that may have been removed from being indexed in the database, but we couldn’t confirm this. For Google Scholar we suspect that it has an inherent limit to the number of articles displayed for long queries excluding citations, as the number of results didn’t change when excluding the year limit from the search phrase. This means that some of the studies we previously filtered would be removed from the newest search query, forcing us to filter its results from scratch.
We have already done this earlier in our research process, as we updated our search from April 2024 in October 2024 to update our review, but due to the limitations we previously mentioned we had to start our filtering process from scratch, which took weeks. As we don’t have enough time during this revision round, and seeing that the newly released articles matching our search query don’t fulfill our inclusion criteria we hope that you allow us to keep our search and filtering strategy in its current form, as updating it wouldn’t impact the results presented in our review.
Comment 4: Line 114: The name and number of the “supplementary file” should be provided, i.e. “in a supplementary file” should be changed to “in supplementary file X”.
Response 4: Based on your recommendation we added the number of supplementary files in the text, and replaced the file names with their order in supplementary materials.
Comment 5: Lines 111-129: The authors have failed to comply with items 5-15 of the PRISMA checklist and the answers given in the last column of the PRISMA checklist (“Location where item is reported”) are inaccurate and do not correspond to the exact location In the text. For example, the authors state that the location in the text for item 5 of the PRISMA checklist is line 92; however, item 5 is a part of Methods, whereas line 92 is a part of the Introduction!
Response 5: The line numbers in the provided checklist are based on the newest version of the manuscript we sent to the editorial office during the previous revision round, where line 92 is part of the “materials and methods section”, specifically referring to the following sentence:
“We included English-written articles published since 2010, discussing IVF in nor-mo-ovulatory human subjects that reported on FF biomarkers such as proteins, growth factors, steroid and polypeptide hormones, inflammation and oxidative stress markers, amino acids, vitamins, lipids of different types, microRNAs (miRNAs), etc., and on IVF/ICSI outcomes such as oocyte count, oocyte quality, fertilization rate, embryo quality, implantation rate, clinical or chemical pregnancy rate, miscarriage rate and live birth rate.”
We did confirm that the line numbers are different in the downloadable version from the journal’s website that shows track changes comparing the old and new versions of the manuscript. We think this is due to the comments in balloons to the side of the manuscript being counted as lines. We will later update the line numbers based on the final version of the manuscript if it hopefully reaches the publish stage to match the lines in the manuscript. Still the numbers of the sections are provided for each item. To clear up any confusion we will list below the quotes, tables, figures or files that we believe cover each of the items 5 through 15:
- Item 5: “We included English-written articles published since 2010, discussing IVF in normo-ovulatory human subjects that reported on FF biomarkers such as proteins, growth factors, steroid and polypeptide hormones, inflammation and oxidative stress markers, amino acids, vitamins, lipids of different types, microRNAs (miRNAs), etc., and on IVF/ICSI outcomes such as oocyte count, oocyte quality, fertilization rate, embryo quality, implantation rate, clinical or chemical pregnancy rate, miscarriage rate and live birth rate.” AND “Table 1 demonstrates the biomarkers included, categorized into nine main types, including the number of studies discussing each type.” AND Table 1 (shows the grouping of studies based on FF biomarker categories).
- Item 6: “Research articles were extracted from two databases (PubMed, Scopus) and the search engine Google Scholar up to October 1st 2024, by creating separate, yet similar, search phrases for each database/ search engine using keywords, operators and search parameters - provided in supplementary file S3.”
- Item 7: “Research articles were extracted from two databases (PubMed, Scopus) and the search engine Google Scholar up to October 1st 2024, by creating separate, yet similar, search phrases for each database/ search engine using keywords, operators and search parameters - provided in supplementary file S3.” AND Supplementary file S3 (search phrases).
- Item 8: “The filtering and record retrieval steps involved all reviewers.” AND Figure 1 (PRISMA 2020 flow diagram).
- Item 9: “Data was then extracted from eligible articles by another two independent researchers and summarized in tables.”
- Item 10a: “Extracted information included: study design, population characteristics, measured follicular fluid biomarkers, IVF/ICSI outcomes and the findings of each study, including statistical measures such as p-values.” AND “Our review aims to establish the possible role of FF constituents in predicting IVF/ICSI outcomes in normo-ovulatory women, including oocyte count, oocyte quality, fertilization rate, embryo quality, implantation rate, clinical or chemical pregnancy rate, miscarriage rate and live birth rate.” AND Table 2 (shows the IVF/ICSI outcomes discussed in each study).
- Item 10b: “More details on each category of biomarkers and protocols followed in each study in Supplementary Tables 1 through 9 in supplementary file S6.” AND Supplementary file S6 (supplementary tables).
- Item 11: “The Newcastle-Ottawa Scale (NOS) was used to assess the quality and risk of bias of included studies after filtering by two independent researchers.”
- Item 12: “including statistical measures such as p-values.”
- Item 13a: Tables 1 and 2 (presenting FF biomarker categories and characteristics for each study).
- Item 13b: “The effects of the aforementioned FF biomarkers on IVF/ICSI outcomes are summarized in Table 3.” AND “These relationships and effects are summarized and visualized in Figure 2” AND Supplementary file S6 (supplementary tables).
- Item 13c: Table 3 (summarizing results for all significant FF biomarkers) AND Figure 2 (visual summary of relationships between FF biomarkers) AND Supplementary file S6 (supplementary tables).
- Item 14: “The Newcastle-Ottawa Scale (NOS) was used to assess the quality and risk of bias of included studies after filtering by two independent researchers.”
- As for items 13d, 13e, 13f and 15, they were non-applicable due to data heterogeneity, which is due to the limited number of studies describing each FF biomarker, which hinders our ability to perform a meta-analysis or a sensitivity analysis. This limitation is explained in Section 5 "Limitations”. We also added a note of this at the end of the PRISMA 2020 checklist.
Comment 6: Page 4, Flowchart: The two parts of the flowchart, i.e. data regarding PubMed and Scopus and data regarding Google Scholar should be better merged. Why did the authors classify Google Scholar under “other methods” rather than “databases and registers”?
Response 6: As we mentioned in our previous response, Google Scholar isn’t considered a database or register, because it differs in the way it stores and retrieves research articles. It functions similarly to Google Search, but only shows results from journal websites. It doesn’t index articles like databases, but retrieves them for each search query, which can give inconsistent results depending on the time of retrieval. It also ranks search results based on an algorithm, not only on them matching the search query, which may lead it to excluding less cited articles that may match the query. Its “about” page states:
“How are documents ranked? Google Scholar aims to rank documents the way researchers do, weighing the full text of each document, where it was published, who it was written by, as well as how often and how recently it has been cited in other scholarly literature.” [https://scholar.google.com/intl/en/scholar/about.html]
In our experience it also fails in handling a large number of search parameters and operators, which forced us to use a less specific search strategy, resulting in a larger number of articles that required filtering. An example of this is mentioned in Figure 1 and the search phrases supplementary file, where we specified the publish year to be 2010 or newer, but Google Scholar still retrieved older articles that needed to be filtered out. For these reasons we opted to use the second type of the PRISMA 2020 flow diagram.
Below are links to two web articles and quotes from them on why Google Scholar can’t be considered a database:
“Based on these criteria, we conclude that Google Scholar is an academic search engine, rather than a database. Since it doesn’t use stable document identifiers, there’s no guarantee that the results you see in one search can be replicated exactly in the future.” [https://paperpile.com/g/google-scholar-database-or-search-engine/]
“Google Scholar has an Advanced search function, however, much like Google, it is a Web Search engine, not a Library Database. Google Scholar may search through Academic sources, but it still uses the search methodology of Crawling and Indexing, not expert Cataloguing.” [https://ecu.au.libguides.com/search-engines/google-scholar]
Comment 7: Results section: Item 21 of the PRISMA statement, i.e. risk of bias should be reported.
Response 7: We believe this item is covered by the inclusion of the Newcastle-Ottawa Scale (NOS) and its supplementary files. We provided a detailed scoring for each study in supplementary files. A summary of the scoring is also mentioned in section 3.1 in the following sentences: “All 22 studies passed the risk of bias assessment using the NOS scale, with cohort studies scoring ≥ 7 points and case control scoring ≥ 8 points out of 9. Detailed scoring of each study is provided in supplementary files S4 and S5.”
Comment 8: Pages 11-25, Lines 599-3433: The Discussion section is now shorter, yet it remains unnecessarily long and it should be shortened considerably (e.g. to a maximum of 3-4 pages), as such a long text would me more appropriate for a thesis, rather than an article published in a journal. The authors should try to provide a concise “general interpretation of the results in the context of other evidence”, according to the PRISMA statement item 23a. Furthermore, the authors should focus on clinically relevant information.
Response 8: We tried as much as we could to shorten the discussion in the previous revision, reducing the word count of the results and discussion by almost 5000 words, but as we mentioned in our previous response the length of the manuscript wasn’t intentional, but a result of the large number of significant biomarkers across different categories included in our review. We tried to cluster the biomarkers in the discussion as much as possible, but this was very difficult because there weren’t many apparent relationships between biomarkers of the same category. We believe the information included in the discussion to be necessary and relevant to explain the results and highlight the various relationships between different FF biomarkers across various cellular pathways that allowed us to design Figure 2, which is the culmination of our work on this review. Shortening the discussion to the suggested length would require the removal of around 5000 words once more, which is sure to remove necessary context, and equate to omission of information. With the current length of the manuscript we are still within the word count suggested by the journal guidelines, which is less than 12,000 words (excluding tables, figures and references).
We would also like more clarification regarding focusing on clinically relevant information, as we believe this to be covered in the discussion of our review, where we try to explain how each FF biomarker may be linked the outcomes discussed in each study, how different FF biomarkers are connected to each other, different methods for measuring those biomarkers in the FF and possible clinical applications based on the discussed information. We see all of the aforementioned points to be clinically relevant to the context of our review, but we are open for any suggestions that may improve our review in this aspect.
Comment 9: Page 25, Conclusions section: Expressions such as “we believe” (line 3435 and 3451) and “we hope” (line 3438 and 3452) should be avoided. Authors should provide evidence and firm evidence-based conclusions rather than beliefs or hopes. The authors should better focus on clinically relevant outcomes. Is there any particular biomarker worthy to be further investigated in the future? Does any biomarker appear to be more important than others?
Response 9: Based on your recommendation we removed phrases such as “we believe”, “we hope” and “hopefully” from the conclusion. Although these statements were supported by the results and discussion, they were meant to carry our hopes and beliefs for future research to confirm and expand on the discussed information, which is one of the goals of our review. We hope this change has made the conclusion clearer and more straightforward in summarizing our findings.
As for choosing the most important biomarker, we mentioned in our previous response, the low number of studies on each biomarker prevents us from pointing out which of these biomarkers may be the most promising diagnostically or therapeutically, especially with the many connections that exist between FF biomarkers. This may also negate the aim of our review to establish a wide and representative metabolome that illustrates how FF biomarkers can affect various cellular pathways as well as each other. We also fear that choosing a certain biomarker or category based on the current limited body of evidence on each biomarker may introduce bias or be misleading for readers or future researchers. And including the significance of different biomarkers or categories of biomarkers in the conclusion may significantly increase its length.
For example, from Figure 2 we may infer that steroid hormones are the most directly linked to clinical outcomes during implantation, while vitamin D and IL-6 seem to have a lot of connections to different biomarkers and cellular pathways. Amino acids, fatty acids and antioxidants may also be candidates due to their potential therapeutic applications in in vitro supplementation (IVS). We also discussed how miRNA can have both diagnostic and therapeutic applications given that future research expands on the miRNAs included in our review and their links to different cellular pathways.
Reviewer 3 Report
Comments and Suggestions for Authors
No more comments
Author Response
No more comments
Response: thank you, we appreciate your former advice
Round 3
Reviewer 2 Report
Comments and Suggestions for Authors
This paper still needs minor revision according to the following comments.
- Abstract, Results: The authors should provide the range of the number of participants in the included studies (The authors responded to a previous comment that “it would be impossible to list the number of participants from all studies in the abstract”; this is reasonable, but the range would suffice).
-Section 2.1: The authors were asked to update the literature search beyond “October 1, 2024” and answered that they conducted an additional search that did not yield any other papers. Hence, they should add a short comment in subsection 2.1; e.g. “An additional search was conducted on March X, 2025 etc. ”
- Flowchart: The authors have answered appropriately to the comment that “the two parts of the flowchart, i.e. data regarding PubMed and Scopus and data regarding Google Scholar should be better merged”. Now, they should add a relevant short comment in the text, in subsection 3.1 (in practice, they should include part of their response to reviewer comments).
- Page 25, Conclusions section: Both reviewers have made comments on this section. The Conclusions have been now improved, but a new comment should be added based on the authors’ response that they “ believe it’s still too early specify certain biomarkers as being the most promising or the most important, due to the low number of studies on each biomarker”. For example: Based on this systematic review of the literature, we could not identify any of the biomarkers as most promising or most important for clinical application in the near future.
Author Response
Dear editor and esteemed peers,
We hope this letter finds you well
We thank you all yet again for your input on our review. The comments included in the latest revision were very helpful in making our methodology more thorough and more transparent for readers. We will include with this letter an updated version of the manuscript and its supplementary files, as well as a “Track changes” file illustrating the changes we made to the manuscript.
The following are responses to Reviewer 2’s comments detailing the changes done to the manuscript based on their recommendations. We hope that these changes are satisfactory. We also performed minor grammatical edits to the manuscript that we previously missed, and can be seen in the “Track changes” file.
Comment 1: Abstract, Results: The authors should provide the range of the number of participants in the included studies (The authors responded to a previous comment that “it would be impossible to list the number of participants from all studies in the abstract”; this is reasonable, but the range would suffice).
Response 1: Based on your recommendation we edited the beginning of the abstract’s results to become: “(3) Results: 22 included articles, with a sample size range of 31 to 414 and a median of 60 participants, contained 61 biomarkers,”. We also added this sentence to subsection 3.1 study selection: “The sample sizes of included studies ranged from 31 to 414 participants, with a median of 60 participants.”
Comment 2: Section 2.1: The authors were asked to update the literature search beyond “October 1, 2024” and answered that they conducted an additional search that did not yield any other papers. Hence, they should add a short comment in subsection 2.1; e.g. “An additional search was conducted on March X, 2025 etc. ”
Response 2: We added this sentence to subsection 2.2 after detailing the inclusion and exclusion criteria: “An updated search query was also performed on March 6th 2025, but didn’t yield any additional studies that fulfilled the inclusion criteria.”
Comment 3: Flowchart: The authors have answered appropriately to the comment that “the two parts of the flowchart, i.e. data regarding PubMed and Scopus and data regarding Google Scholar should be better merged”. Now, they should add a relevant short comment in the text, in subsection 3.1 (in practice, they should include part of their response to reviewer comments).
Response 3: We added this sentence to subsection 3.1 after detailing the results of search in each database/ search engine: “Since Google Scholar is a search engine and not a database like PubMed and Scopus, we opted to use the PRISMA 2020 flow diagram for databases, registers and other sources to better illustrate this difference (Figure 1) [12].”. We hope that this short comment is sufficient, even though it doesn’t detail the reasons for considering Google Scholar a search engine included in our previous response.
Comment 4: Page 25, Conclusions section: Both reviewers have made comments on this section. The Conclusions have been now improved, but a new comment should be added based on the authors’ response that they “ believe it’s still too early specify certain biomarkers as being the most promising or the most important, due to the low number of studies on each biomarker”. For example: Based on this systematic review of the literature, we could not identify any of the biomarkers as most promising or most important for clinical application in the near future.
Response 4: We added this sentence near the end of the conclusion section: “It’s still too early to identify the most promising FF biomarkers in terms of clinical diagnostic and therapeutic utility out of the ones included in our review, due to the low number of studies on each biomarker.”. We also edited the abstract’s conclusions in accordance to become: “(4) Conclusions: FF biomarkers can be utilized for diagnostic and therapeutic purposes in IVF; however further studies are required for choosing the most promising ones due to heterogeneity of results.”
We thank you for your time in reading this letter, and we await your response regarding its content.